# Seasonal evolution of the effective thermal conductivity of the snow and the soil in high Arctic herb tundra at Bylot Island, Canada

**Florent Domine[1,2,3,4], Mathieu Barrere[1,3,4,5,6], Denis Sarrazin[3]**

[1]Takuvik Joint International Laboratory, Université Laval (Canada) and CNRS-INSU (France), Pavillon Alexandre Vachon, 1045 avenue de La Médecine, Québec, QC, G1V 0A6, Canada

[2] Department of Chemistry, Université Laval, Québec, QC, Canada

[3]Centre d'Études Nordiques, Université Laval, Québec, QC, Canada

[4]Department of Geography, Université Laval, Québec, QC, Canada

[5]Météo-France – CNRS, CNRM UMR 3589, CEN, Grenoble, France

[6]LGGE, CNRS-UJF, Grenoble, France

*Correspondence to:* F. Domine (florent.domine@gmail.com)

**Abstract.** The values of the snow and soil thermal conductivity, $k_{snow}$ and $k_{soil}$, strongly impact the thermal regime of the ground in the Arctic, but very few data are available to test model predictions for these variables. We have monitored $k_{snow}$ and $k_{soil}$ using heated needle probes at Bylot Island in the Canadian High Arctic (73°N, 80°W) between July 2013 and July 2015. Few $k_{snow}$ data were obtained during the 2013-2014 winter, because little snow was present. During the 2014-2015 winter $k_{snow}$ monitoring at 2, 12 and 22 cm heights and field observations show that a depth hoar layer with $k_{snow}$ around 0.02 W m$^{-1}$ K$^{-1}$ rapidly formed. At 12 and 22 cm, wind slabs with $k_{snow}$ around 0.2 to 0.3 W m$^{-1}$ K$^{-1}$ formed. The monitoring of $k_{soil}$ at 10 cm depth shows that in thawed soil, $k_{soil}$ was around 0.7 W m$^{-1}$ K$^{-1}$ while in frozen soil it was around 1.9 W m$^{-1}$ K$^{-1}$. The transition between both values took place within a few days, so that the use of a bimodal distribution of $k_{soil}$ for modelling appears adequate, in contrast to conclusions from previous studies. This may be explained by different soil properties or by artefacts caused by using high heating powers for thermal measurements in previous works. Simulations of $k_{snow}$ using the snow physics model Crocus were performed. Contrary to observations, Crocus predicts high $k_{snow}$ values at the base of the snowpack (0.12 to 0.27 W m$^{-1}$ K$^{-1}$) and low ones in its upper parts (0.02 to 0.12 W m$^{-1}$ K$^{-1}$). We diagnose that this is because Crocus does not describe the large upward water vapor fluxes caused by the temperature gradient in the snow and soil. These fluxes produce mass transfer between the soil and lower snow layers to the upper snow layers and the atmosphere. Finally, we discuss the importance of the structure and properties of the Arctic snowpack on subnivean life, as species such as lemmings live under the snow most of the year and must travel in the lower snow layer in search of food.

**Keywords.** Snow, Soil, Thermal conductivity, Arctic, Permafrost, Subnivean life

**1 Introduction**

Arctic permafrost contains large amounts of frozen organic matter (Hugelius et al., 2014). Its thawing could lead to the microbial mineralization of a fraction of this carbon, resulting in the release of yet undetermined but potentially very important amounts of greenhouse gases (GHG, $CO_2$ and $CH_4$) to the atmosphere (Elberling et al., 2013; Schneider von Deimling et al., 2012; Schuur et al., 2015), resulting in a strong positive climate feedback. Predicting GHG release from permafrost first requires the prediction of the evolution of the permafrost thermal regime, which depends to a significant

extent on snow and soil thermal conductivity. Several recent land surface model developments or improvements have indeed tested that these variables had a critical impact on ground temperature (Chadburn et al., 2015; Ekici et al., 2015; Paquin and Sushama, 2015). Snow in particular is often mentioned as a key factor in the permafrost thermal regime (Burke et al., 2013; Gouttevin et al., 2012). Yet, most land surface models which use an elaborate snow scheme often simply parameterize snow thermal conductivity as a non-linear function of its density. Even though the average density of snow may be adequately

predicted in land surface models (Brun et al., 2013), the density profile of Arctic or subarctic snow is currently not predicted well by most or all snow schemes (Domine et al., 2013) because the upward water vapor flux generated by the strong temperature gradients present in these cold snowpacks (Sturm and Benson, 1997) is not taken into account. These fluxes lead to a mass transfer from the lower to the upper snow layers, and these transfers can result in density changes of 100 kg m$^{-3}$, perhaps even more (Domine et al., 2013; Sturm and Benson, 1997). Given the non-linearity between snow thermal

conductivity and density used in most snow schemes, errors in the snow density vertical profile inevitably lead to errors in the snowpack thermal properties and therefore in the permafrost thermal regime.

      Obviously, soil thermal conductivity is also important in determining vertical heat fluxes. For a given soil composition, this variable depends mainly on  variable depends on temperature and content in liquid water and ice, so that its value may show considerable variations over time (Overduin et al., 2006; Penner et al., 1975). Modellers of the permafrost thermal regime

have stressed that "monitoring of this parameter in the active layer all year-round would be useful if a more realistic numerical model is to be developed" (Buteau et al., 2004).

      Another important interest of studying snow physical properties such as thermal conductivity and the heat budget of the ground lies in the understanding of the conditions for sub-nivean life. For example, lemmings live under the snow most of the year at Bylot Island (Bilodeau et al., 2013), and the temperature at the base of the snowpack conditions their energy

expenses to maintain body temperature. Furthermore, energy expense for sub-nivean travel in search of food depends on snow hardness and snow conditions have been invoked to help explain lemming population cycles (Bilodeau et al., 2013; Kausrud et al., 2008), even though no comprehensive snow studies have yet been performed to fully establish links between lemming populations and snow properties. Snow thermal conductivity has been shown to be well correlated with snow mechanical properties (Domine et al., 2011), so that monitoring snow thermal conductivity may help quantify the effort

required by lemmings to access food and hence understand their population dynamics.

      Given the importance of snow thermal conductivity to simulate the ground thermal regime and to understand the conditions for sub-nivean life, we have initiated continuous automatic measurements of the snow thermal conductivity vertical profile at several Arctic sites using heated needle probes (NP). The first instrumented site was in low Arctic shrub tundra (Domine et al., 2015), near Umiujaq on the East shore of Hudson Bay. We present here an additional study with two years of snow

thermal conductivity monitoring at Bylot Island, a high Arctic herb tundra site (73°N, 80°W). Since lemming populations have been monitored there for over two decades (Bilodeau et al., 2013; Fauteux et al., 2015), the snow work undertaken here may indeed also help understand snow-lemming relationships. Additionally, we also placed a heated NP in the soil at a depth of 10 cm to monitor soil thermal conductivity in the active layer over the seasonal freeze-thaw cycles.

## 2 Methods

### 2.1       Study site and instrumentation

Our study site is on Bylot Island, just North of Baffin Island in the Canadian high Arctic. The actual site was at the bottom of Qalikturvik valley, in an area of ice-wedge polygons (N 73°09'01.4", W 80°00'16.6"), in a context of high Arctic thick permafrost. This spot is within 100 m of that described by (Fortier and Allard, 2005). Our instruments were placed in July 2013 in a fairly well drained low-center polygon with peaty silt soil (Figure 1). Vegetation consists of sedges, graminoids and mosses (Bilodeau et al., 2013). The active layer is about 30 cm deep at our site and permafrost at Bylot Island is thought to be several hundred meters deep (Fortier and Allard, 2005). The site is only accessible with complex logistics, so that field work only takes place in late spring and summer.

Three TP08 heated NPs from Hukseflux were positioned on a polyethylene post at heights above the ground of 7, 17 and 27 cm in July 2013. These heights were chosen somewhat arbitrarily before the snowpack structure could be observed. In July 2014 the TP08 needles were lowered to 2, 12 and 22 cm because we had realized during a field trip in May 2014 that the lowermost depth hoar layer could be thinner than 7 cm. On another nearby post, thermistors were placed at heights of 2, 7, 17, 27 and 37 cm. Heights intermediate between the TP08 heights allow the calculation of heat fluxes, using thermal conductivity and temperature values. Unfortunately, the cables of the thermistors at 7, 27 and 37 cm, although protected, were chewed by a fox in late summer 2014 and could only be replaced the following summer. In the ground, we also placed a TP08 NP at a depth of 10 cm with two thermistors at depths of 4.5 and 13.5 cm. In the immediate vicinity of the thermistors post, we placed 5TM sensors from Decagon to monitor ground temperature and volumetric liquid water content at depths of 2, 5, 10 and 15 cm. Water content sensors used the manufacturer's calibration for mineral soils and were not recalibrated, which may produce an error of up to 3%. At that time, it was not possible to place deeper sensors because of the limited thaw depth. A few meters from these snow and ground instruments, we installed meteorological instruments to measure atmospheric variables including air temperature and relative humidity with a HC2S3 sensor from Rotronic, wind speed with a cup anemometer, both at 2.3 m height, snow height with a SR50A acoustic snow height gauge (Campbell Scientific), an IR120 infrared surface temperature sensor and a CNR4 radiometer with a CNF4 heating/ventilating system from Kipp & Zonen which measured downwelling and upwelling shortwave and longwave radiation. Heating and ventilating radiometers is mandatory to limit snow accumulation and the build-up of frost and freezing rain. Heating and ventilation were performed for 5 minutes every hour prior to measurements. This protocol of instrumentation and measurement is similar to that employed near Umiujaq (Domine et al., 2015) and were operated in the same manner with a CR1000 data logger (Campbell Scientific). Briefly, hourly measurements were recorded, except for the TP08 needles, whose operation is described in the next section.

### 2.2       Thermal conductivity measurements

The NP method has been used extensively for soils for a long time (Devries, 1952). (Sturm and Johnson, 1992) and (Morin et al., 2010) discussed in detail the heated NP method in snow. The automatic operation of the TP08 needles in Arctic snow and the data analysis have been detailed in (Domine et al., 2015) and only a brief summary will be given here.

For a measurement cycle, the 10 cm-long needles were heated at constant power (0.4 W m$^{-1}$) for 150 s. The temperature was monitored by a thermocouple at the center of the heated zone. Heat loss is a function of the thermal conductivity of the medium. A plot of the thermocouple temperature as a function of $ln(t)$, where $t$ is time, theoretically yields a linear curve after an initiation period of 10 to 20 s. The slope of the linear part is inversely proportional to the thermal conductivity of the medium.

In snow, NPs in fact measures an effective thermal conductivity, $k_{snow}$, with contributions from conductive heat transfer through the network of interconnected ice crystals and through the interstitial air, and also heat transfer due to latent heat

exchanges caused by water molecules sublimation and condensation generated by the temperature gradient around the heated NP (Calonne et al., 2011; Sturm et al., 1997). The variable $k_{snow}$ is defined as:

$$F = -k_{snow} \frac{dT}{dz}$$

(1),

with $dT/dz$ the vertical temperature gradient through the layer, in K m$^{-1}$, and $F$ the heat flux in W m$^{-2}$.

The automatic routine developed by (Domine et al., 2015) was used to obtain $k_{snow}$ from the heating curve. Essentially, that routine detects possible non-linearity in the heating curve and selects the most linear curve section to determine $k_{snow}$. The main cause of non-linearity is the occurrence of convection in the snow, which, by adding an extra heat transfer process, reduces the rate of temperature rise. This readily happens in depth hoar after 50 to 60 s of heating because of its high permeability. Since convective heat transfer is not an intrinsic snow property because it depends among other things on the heating power, it has to be detected and data affected by it must be discarded. Therefore, while for heating curves not perturbed by convection, the 40-100 s heating time range was used to derive $k_{snow}$, the range 20-50 s was used when convection was present (Domine et al., 2015). Other causes of non-linearity include non-homogeneous snow and melting due to excessive heating. To avoid melting, automatic measurements were started only if the snow temperature was <-2°C. Factors such as non-homogeneous snow may result in poor quality heating curves, which were discarded. Thermal conductivity measurements were performed once every 2 days, to take into account the slow evolution of this variable and to minimize thermal perturbation to the snow. Measurements were performed at 5:00 to minimize the risk of having warm snow. The NP heating sequence takes place regardless of whether the NP is covered by snow or not. The $k_{snow}$ data that we determined took place in air were discarded. When an NP is not snow-covered, even a slight wind produces an erratic heating curve. Should there be very little wind, snow height measurement, the value of $k_{snow}$, and the presence of convection often allows the detection of a non-covered probes. Incorrect or uncertain determination of NP snow coverage is nevertheless still possible, and this will be discussed when relevant.

Measurements of soil thermal conductivity $k_{soil}$ were not performed by (Domine et al., 2015), and details are therefore given here. Since snow and soil NPs are all multiplexed onto the same data logger, the same heating power had to be used in snow and soil. The thermal conductivity of soils, especially when frozen, can be much greater than that of snow (Penner, 1970), so that heating by the NP can be much lower, resulting in more noise in the heating curves. NPs do not induce convection in soils because of their low permeability so that heating curves were always linear. The 30-90 s heating time range was thus used. A quality threshold based on the determination coefficient of the curve was used and curves with R$^2$<0.75 were discarded.

Measuring thermal conductivity in porous granular media such as snow has been suspected of presenting biases and/or systematic errors (Calonne et al., 2011; Riche and Schneebeli, 2013) and these have been discussed by (Domine et al., 2015). One concern is that the relevant metric for heat exchanges between the ground and the atmosphere through the snow is the vertical heat flux. Snow is anisotropic (Calonne et al., 2011; Riche and Schneebeli, 2013) and NPs measure a mixture of vertical and horizontal heat fluxes. Anisotropy depends on snow type, with for example depth hoar being more conductive in the vertical direction and wind slabs in the horizontal direction. Moreover, a systematic error caused by the granular nature of snow was invoked by (Riche and Schneebeli, 2013). (Domine et al., 2015) discussed the impact of these processes on the accuracy of $k_{snow}$ measurement with NPs and concluded that if the snow type was not known, the maximum error was typically 29%. The largest contributions to error were the systematic error due to the use of the NP (20%) and error due to anisotropy (20%). The total error is the square root of the sum of the squares of all contributions. If the snow type is known, corrections for anisotropy and systematic bias are possible, potentially reducing errors to 10 to 20%. Such corrections were not, however, performed by (Domine et al., 2015). Given that $k_{snow}$ varies between 0.025 and 0.7 W m$^{-1}$ K$^{-1}$, errors due to the use of NPs are quite acceptable and measurements using NPs still very useful. Alternative $k_{snow}$ measurement methods, i.e. heat flux plates and calculations based on tomographic images, although potentially more accurate, are clearly impractical for

season-long monitoring in remote and inaccessible high Arctic sites because they require sampling and complex instruments. For soils, the issues raised by (Riche and Schneebeli, 2013) are expected to have no impact because soils are much denser with smaller grains, and are closer to a homogeneous medium at the scale of a NP. Anisotropy may however be an issue in some soil types, but we have not developed this aspect here. Based on the quality of our soil heating curves, we estimate that the error on $k_{soil}$ is 15% or less when the soil is not frozen and 25% when it is frozen, because higher $k_{soil}$ means lower heating by the NP and therefore less accurate heating curves.

Field measurements of snow $k_{snow}$ and soil $k_{soil}$ were also performed during field campaigns in spring 2014 and 2015 and summer 2013, 2014 and 2015 with a TP02 NP. Additional spring field measurements included snow density with a 100 cm³, 3-cm high box cutter, snow temperature and snow specific surface area (SSA) with the DUFISSS instrument based on infrared reflectance at 1310 nm (Gallet et al., 2009). SSA is the surface area per mass and is thus inversely proportional to snow grain size. SSA can be used to evaluate the intensity of metamorphism, as intense metamorphism driven by high temperature gradients lead to rapid grain growth and hence SSA decrease (Taillandier et al., 2007). Some of these snow measurements have been detailed by (Domine et al., 2016). Additional summer measurement included soil volume water content using an EC5 sensor from Decagon.

### 2.3 Soil granulometric analysis

Soil grain size distribution is useful to understand its physical properties. Samples were taken from our instrumented site at 10 cm depth in July 2015. Particle size distribution was measured with a Horiba partica LA-950V2 laser scattering particle size analyser, which used a 2-wavelength optical system, 405 and 650 nm. Five subsamples were analyzed and averaged.

### 2.4 Snow physics modelling

We used the Crocus model (Vionnet et al., 2012) to simulate snow physical properties. We in fact used the simulations already described in (Domine et al., 2016) for our Bylot Island site, but analyzed different output data. Very briefly, the model was forced with our meteorological data. When data were missing, ERA-interim reanalysis data were used (Dee et al., 2011), corrected following the procedure of (Vuichard and Papale, 2015) to minimize the bias between measured and ERA data. Snow thermal conductivity is calculated from the equation of (Yen, 1981), based on a correlation between density and thermal conductivity.

### 3 Results

The winter 2013-2014 was a low snow year so that two out of three NPs were not covered. Data are more complete for the following year and we therefore start with data from the 2014-2015 season.

### 3.1 2014-2015 season

Describing the structure and physical properties of the snowpack at Bylot Island helps in understanding the monitoring data. Observations made on 12 May 2015 close to our monitoring site are shown in Figure 2. Vegetation was observed to be mostly flattened by snow, with some sedge or graminoids stems still upright, but they did not seem to have impacted snow structure. We observed a basal depth hoar layer 8 to 10 cm thick, overlaid by a wind slab 11 to 12 cm thick, with in-between a thin layer of faceting crystals. Above the wind slab were thin layers of small rounded grains, decomposing precipitation crystals and a thin wind crust. The depth hoar layer was divided into two sublayers. The lower sublayer was slightly indurated and harder, although hardness and other properties appeared spatially very variable. Indurated depth hoar is a snow type seldom mentioned, as it does not form in Alpine or temperate snow, and it is not described in the international snow

classification (Fierz et al., 2009). It has been observed in the Arctic without being named for decades. Its most detailed description is probably that of (Derksen et al., 2009): " Wind slabs formed in early winter often completely metamorphose into depth hoar by the end of the season as a result of these temperature gradients […]. The grains in these layers will be morphologically similar to regular depth hoar, but the layers will be stronger and more cohesive than normal for depth hoar layers, a relict feature of the original wind slab. We call these layers indurated." (Domine et al., 2016) gave complementary details: "Indurated depth hoar […] forms in dense wind slabs under very high temperature gradients not encountered in Alpine snow. Its density can exceed 400 kg m$^{-3}$. Large depth hoar crystals coexist with small grains that have not been subject to fast crystal growth, probably because water vapour vertical transfer has followed preferential paths in the dense snow. This often gives indurated depth hoar a milky aspect. Unlike typical depth hoar (e.g. taiga depth hoar, […]) which has a very low cohesion, indurated depth hoar is reasonably cohesive, although fairly brittle, and large blocks can easily be cut out of it and manipulated."

The indurated depth hoar here showed signs of early season melting in the form of rounded grains that were partially to almost totally transformed into depth hoar, but bonds were stronger than for regular depth hoar. The indurated depth hoar that forms in refrozen snow is slightly different from that observed in wind slabs, in that no small grains are present and it does not have a milky aspect. In line with (Domine et al., 2016), we represent this type of depth hoar with a symbol that does not exist in the classification of (Fierz et al., 2009), as that classification is ideal for Alpine snow but is not detailed enough to represent many Arctic snow types. The depth hoar upper sublayer was very soft, appeared more homogeneous and showed no signs of melting. Vertical profiles of density, SSA and thermal conductivity are also shown in Figure 2. Several density measurements were made in the lower depth hoar layer, yielding values between 172 and 260 kg m$^{-3}$. In comparison, the upper depth hoar layer was at 170 kg m$^{-3}$, a rather low value for depth hoar. The vertical profiles of the three measured physical variables reflect the observed stratigraphy rather well, with higher values of all three variables for the wind slab and lower values for the depth hoar, as already noted by (Domine et al., 2016). The basal snow temperature was -17.1°C.

Figure 3 shows time series for $k_{snow}$, as well as snow temperature, air temperature, wind speed and snow height. Snow height is spatially very variable in the Arctic at the 50 cm scale, mostly because of wind effects and of microtopography. Typically, in May around our site, snow height varied between 22 and 45 cm within 5 m. Measurements using an avalanche probe at 236 spots within 200 m of our site on 12 May 2015 showed a mean snow height of 25.3 cm, with a standard deviation of 13.1 cm (Domine et al., 2016). Therefore, snow height given by our acoustic snow gauge can only be taken as an indication of the actual snow height at the very NP site, about 5 m away. On 12 May 2015, snow height was 27 cm at the pit shown in Figure 2, 50 cm from the NP post, while the snow gauge indicated 35 cm.

The lowest NP, at 2 cm, was covered by the first snow fall on 12 September 2014. That initial snow partially melted and the first data point that was almost certainly obtained in snow was on 24 September. After initial values around 0.04 W m$^{-1}$ K$^{-1}$, $k_{snow}$ dropped to values around 0.02 W m$^{-1}$ K$^{-1}$ because of rapid depth hoar formation. These values may seem a bit low, especially considering that air has a thermal conductivity of 0.023 W m$^{-1}$ K$^{-1}$, but the low value can be attributed to a negative systematic error of about 20% caused by the NP method, as described in (Riche and Schneebeli, 2013) and discussed above. It is fairly certain that this NP was not in an air gap, because measurements in air almost always produce erratic heating curves due to complex convection, and none of the heating curves were suspicious. Another reason for the low value is that depth hoar is probably the most anisotropic snow, with the vertical component of thermal conductivity being the greatest one (Calonne et al., 2011; Riche and Schneebeli, 2013). Since we measure a mixture of both vertical and horizontal component, the value is necessarily lower than the vertical component., Ttaking into account all error sources, the actual vertical component is probably 25 to 30% greater than the value obtained from the NP.

The NP at 12 cm may have been in snow on 1 November, but this is uncertain. On 10 to 11 November, the strongest wind storm of the winter, which reached wind speeds of 12.9 m s$^{-1}$, appears to have formed a wind slab that buried the 12 cm NP, which recorded a $k_{snow}$ value of 0.28 W m$^{-1}$ K$^{-1}$. The $k_{snow}$ value at 12 cm remained close to that value until 28 March. Until 1

January, data are noisy, possibly because the NP was close to the snow surface and therefore subject to air advection by wind, which, by adding a heat loss process, produced randomly a slight positive artifact. The 12 cm $k_{snow}$ value then dropped to 0.23 W m$^{-1}$ K$^{-1}$ on 30 March, while no special meteorological event was recorded by the wind and temperature sensors. Between 27 and 30 March, the wind remained under 4 m s$^{-1}$ and the temperature showed regular diurnal cycles between -40 and -20°C. The value of $k_{snow}$ started rising steadily on 5 May to reach a value of 0.33 W m$^{-1}$ K$^{-1}$ on 29 May. Considerations of the evolution of metamorphic conditions in the snowpack, discussed subsequently, are required to explain this process.

The NP at 22 cm was definitely covered by snow on 21 January 2015. The value remained around 0.06 until 8 February, indicating that it remained as undisturbed precipitation. On that date, a moderate wind storm started, reaching 5.6 m s$^{-1}$ at 11:00. The resulting wind slab had $k_{snow}$ =0.175 W m$^{-1}$ K$^{-1}$, increasing steadily to 0.25 W m$^{-1}$ K$^{-1}$ on 23 April, most likely because of slow sintering. On 25 April, the value rose to 0.309 W m$^{-1}$ K$^{-1}$, but this is related to no wind event. On 17 April, a 44 h-long wind event culminating at 6.8 m s$^{-1}$ did not produce any rise in $k_{snow}$, indicating that the 22 cm needle was sufficiently covered to be unaffected by wind. Air temperature was no different from other nearby days. The heating curve is slightly curved, which could have been caused by an unsteady heating current. In any case, we attribute this short temporary rise to noise in the data. Except for this rise, $k_{snow}$ stabilized around 0.26 W m$^{-1}$ K$^{-1}$ until the end of May, which we will discuss subsequently in light of metamorphic conditions.

Three soil pits were dug in the summers 2013 to 2015 down to the thaw front in the polygon where our instruments are located to measure soil physical properties and another two pits were dug just for observations. An organic litter layer 3.5 to 6 cm thick was observed. Lower down was a layer of organic-rich silt-looking material was observed. Figure 4 shows vertical profile of soil temperature, thermal conductivity, $k_{soil}$, and volume water content fraction. The general trend is an increase of $k_{soil}$ with depth, while temperature expectedly decreases. Soil grain size distribution was obtained from 5 samples taken around 10 cm depth. The average data show a bimodal size distribution with modes centered at 17 and 59 µm. If the standard 50 µm size limit between sand and silt is used, then our sample is 65% silt and 35% sand by mass, so that the soil here is a mixture of silt and fine sand.

Figure 5 shows the evolution of the $k_{soil}$ and temperature measured every other day by the NP at 10 cm depth between 28 July 2014 and 26 June 2015. Figure 5 also shows hourly soil temperature and volume water content at 10 cm depth measured by a 5TM probe located about 1 m away. Both temperature measurements show similar variations, but the 5TM is 1 to 2°C warmer, in part due to a positive 0.5°C offset on the 5TM. The soil temperature reached 0°C on 9 September and stayed at that temperature without significant freezing, as indicated by the water content, until 27 September. The soil water freezing continued until 10 October, at which point only water in small pores remained liquid. Until 30 September 2014, the soil thermal conductivity stayed constant with $k_{soil}$ = 0.73 W m$^{-1}$ K$^{-1}$, with a standard deviation of 0.02 W m$^{-1}$ K$^{-1}$, while the soil volume water content was 49.5±1.3%. Except for very low water contents, soil freezing is expected to manifest itself in an increase in $k_{soil}$, (Inaba, 1983; Penner, 1970) because ice has a much higher thermal conductivity than water (2.22 vs. 0.56 W m$^{-1}$ K$^{-1}$ at 0°C). Measurements of $k_{soil}$ shows detectable soil freezing between 30 September and 2 October, with $k_{soil}$ increasing from 0.76 to 1.16 W m$^{-1}$ K$^{-1}$. The value of $k_{soil}$ then rapidly increased to 1.8 Wm$^{-1}$K$^{-1}$ on 10 October, confirming 5TM data that the soil was then almost essentially frozen at 10 cm depth, except for water in small pores (Inaba, 1983; Penner, 1970). The average and standard deviation of $k_{soil}$ then were 1.95 ± 0.20 W m$^{-1}$ K$^{-1}$ for the winter season. The large standard deviation only indicates the greater uncertainty of our instrument for high $k_{soil}$ values, because NP heating is less pronounced. The values of both unfrozen and frozen soil are consistent with those expected from a fine grain mineral material mixed with organic matter (Kujala et al., 2008; Penner, 1970). Even as the soil temperature decreases to -30°C, no further increase in $k_{soil}$ is observed, suggesting that essentially all the water that can was already frozen on 10 October. The value of $k_{soil}$ then only decreased, and did so in just a few days, when thawing took place on 20 June 2015 and $k_{soil}$ returned to its previous thawed value. Figure 5 shows that the increase in $k_{soil}$ upon freezing actually takes place after the initiation of freezing and that its decrease in spring is complete before full thawing. In the case measured here, a small ice fraction

produces a value of $k_{soil}$ as if all the water was liquid.

**3.2 2013-2014 season**

Figure 6 shows the value of $k_{snow}$ at a height of 7 cm for the 2013-2014 season, as only that NP was sufficiently covered to give reliable data. As previously, we also show snow temperature and height, wind speed and air temperature. That season, the snow height at our snow gauge was noticeably lower than at our NP post. On 14 May 2014, the gauge indicated 13 cm, while there was 18 cm of snow at the NP post. Measurements using an avalanche probe at 314 spots within 200 m of our site on 14 May 2015 showed a mean snow height of 16.2 cm, with a standard deviation of 13.7 cm (Domine et al., 2016).

The snow stratigraphy was observed on 14 May 2014 about 50 cm from the NP post and vertical profiles of density, specific surface area and thermal conductivity were measured and are shown in Figure 7. The stratigraphy was spatially extremely variable and complex with frequent alternation of hard and soft layers. The basal layer of columnar depth hoar was very soft and collapsed at the slightest contact so that we were not able to measure its density. Gaps in the basal layer were frequent, indicating that it had collapsed naturally in many places. These spontaneous collapse features cannot be mistaken for lemming burrows (Figure 8). This very soft and fragile structure was most likely due to the loss of matter caused by the upward water vapor flux generated by the temperature gradient in the snowpack. Such basal depth hoar collapse in the low and high Arctic have already been described by (Domine et al., 2015) and (Domine et al., 2016). Above the basal depth hoar was a layer of indurated depth hoar formed by the metamorphism of a hard wind slab into depth hoar, as detailed above.. Although brittle, the indurated depth hoar observed was fairly solid and could readily be sampled without damaging its structure. It has a high thermal conductivity (0.37 W m$^{-1}$ K$^{-1}$), high density (383 kg m$^{-3}$) and a SSA (14.6 to 19.6 m$^2$ kg$^{-1}$) slightly lower than most wind slabs but higher than typical depth hoar. No symbol for indurated depth hoar formed in wind slabs exists in the classification of (Fierz et al., 2009). (Hall et al., 1991) used open squares for "solid-type depth hoar", presumably indurated depth hoar. (Sturm et al., 2002) used a mixtures of wind slab and depth hoar symbols for "wind slab to depth hoar" presumably also indurated depth hoar. These examples show that the need for a specific indurated depth hoar symbol exists. Since the symbol we proposed earlier for indurated depth hoar formed in refrozen snow consists of a depth hoar symbol with a large open circle, we propose to use a depth hoar symbol with a dot (the fine grain symbol) for indurated symbol formed in wind slabs.

The intermediate layer of faceted crystals around 10 cm indicates an extended period of low wind weather, during which temperature gradient metamorphism could proceed without perturbation by any wind compaction episode. Few precipitation events took place that winter, as indicated by the small amount of snow observed in May 2014. The snow gauge (Figure 6) also indicates little precipitation, although many wind-erosion episodes at our gauge spot limit our ability to evaluate precipitation in 2013-2014.

Snow cover in 2013 started late, on 12 October. Our NP at 7 cm recorded a first significant $k_{snow}$ increase, from 0.050 to 0.094, between 7 and 9 November, and we attribute this to a wind event that lasted about 14 hours on 7 November, with wind speed reaching 7 m s$^{-1}$, and which must have formed a wind slab of moderate density and thermal conductivity around 0.1 W m$^{-1}$ K$^{-1}$, which remained stable for a few days before increasing to 0.178 W m$^{-1}$ K$^{-1}$ between 17 and 21 November. This is well correlated to 2 consecutive wind events, each lasting over 24 hours on 18 and 21 November, and which respectively reached 8.8 and 9.4 m s$^{-1}$. We propose that erosion and redeposition of snow took place, leading to the formation of a denser layer that rapidly sintered. On 23 November, $k_{snow}$ decreased to 0.077 W m$^{-1}$ K$^{-1}$, which we attribute to wind erosion and subsequent redeposition of softer snow of low $k_{snow}$, as the 21 November storm lasted until noon, while the measurement took place at 5:00. The next observed rise in $k_{snow}$ was on 3 December to 0.156 and then to 0.192 W m$^{-1}$ K$^{-1}$ on 5 December. The rise between 29 November and 3 December was caused by a 48 hour wind storm that reached 9.3 m s$^{-1}$, making the 1 December heating plot unreliable. The subsequent rise is attributed to another storm on 2-3 December that reached 8.1 m s$^{-1}$. Variations in $k_{snow}$ were thereafter small. The NP was certainly buried several cm below the surface and little more affected

directly by wind. Factors which can affect the value of $k_{snow}$ include temperature, through its effect on the thermal conductivity of ice (which increases as temperature decreases), wind which through wind pumping may add another heat transfer process and produce a positive artefact on $k_{snow}$, changes in snow structure due to metamorphism, and simply noise in the data. Deconvolution of all these effects, given their small impact and the presence of noise in the data, appears of little interest. It is nevertheless noteworthy that the drop from 0.238 to 0.193 W m$^{-1}$ K$^{-1}$ on 8 March coincides with a wind event on 7 March (6.2 m s$^{-1}$). We speculate that this may have caused wind pumping leading to sublimation, mass loss and a drop in $k_{snow}$.

On 14 May 2014, we excavated the snow around the NPs, essentially ending our time series. A photograph of the snow profile is shown in Figure 9. The NP at 7 cm was in an indurated depth hoar layer, but very close to the border with a thin depth hoar layer. Above that was a layer of faceted crystals/depth hoar. Given the stratigraphy, the 7 cm NP had been completely buried for months and changes in $k_{snow}$ for the past few months cannot be interpreted in terms of precipitation/erosion processes.

Figure 10 shows soil data for the 2013-2014 season. Before the initiation of freezing, the soil volume water content at 10 cm depth was 56.6±0.8%. The soil temperature at 10 cm depth reached 0°C on 6 September. Freezing was very slow until 12 September, when the water content started showing a detectable decrease. Most of the water was frozen on 1 October and the temperature started to drop. Until 20 September, $k_{soil}$ was essentially constant, with $k_{soil}$ = 0.71±0.04 W m$^{-1}$ K$^{-1}$. This is not significantly different from the following season, while the volume water content is slightly larger (56.6 vs. 49.5%). Again, here a small ice fraction does not seem to affect the value of $k_{soil}$, which remained similar to that in fully thawed conditions.

## 4 Discussion

### 4.1 Snow metamorphism and water vapor fluxes

The snowpack structure observed at Bylot Island, especially in 2015, is frequently encountered on Arctic tundra (Benson et al., 1993; Derksen et al., 2009; Domine et al., 2002; Sturm and Benson, 2004), especially in areas of moderate wind, and mostly consists of a lower depth hoar layer and an upper wind slab. The depth hoar layer forms because of the elevated temperature gradient at the beginning of the season. Figure 11 shows the temperature gradients in the 2-12 cm and 12-22 cm snow height ranges, as obtained from the NPs temperature measurements every other day at 5:00. Higher time resolution measurements would have been desirable, but most of the thermistors that logged temperature every hour were damaged by a fox. Figure 11 nevertheless shows that NP data are similar to the gradient derived from thermistor data at 2 and 17 cm, so that reasoning on NP data is still adequate. Values barely reach 100 K m$^{-1}$, while other Arctic or subarctic locations showed early season values in the 200-300 K m$^{-1}$ range (Sturm and Benson, 1997; Taillandier et al., 2006). Reasons for the lower values reported here include: (i) we only obtained values starting on 1 November because our thermistor at 7 cm did not function and we had to wait for the 12 cm NP to be covered to obtain data. Early season values, when the snowpack was thinner and the ground not completely frozen, were almost certainly higher; (ii) sites with higher gradients were inland sites and atmospheric cooling was faster, reaching -35°C in November (Taillandier et al., 2006). Here, the presence of the sea and the latent heat released by sea water freezing led to a much slower and gradual cooling, as shown in Figures 3 and 6. Figure 11 shows that the temperature gradient in both height ranges are pretty similar until 17 March. Between 15 and 17 March, the air temperature rose from -37 to -10.5°C (Figure 3), and that brutal and irreversible warming dramatically changed the thermal regime of the snowpack, as seen in Figure 11, where the gradient in the upper regions suddenly becomes much greater than in the lower one.

With regards to metamorphism, the actual variable of interest is the water vapor flux, rather than just the temperature gradient. This flux is the product of the diffusion coefficient of water vapor in snow, D$_v$, by the water vapor concentration

gradient. $D_v$ as reported by (Calonne et al., 2014) depends on snow density and we estimate that it was $2x10^{-5}$ m$^2$ s$^{-1}$ below 12 cm height because of the presence of depth hoar and $1x10^{-5}$ m$^2$ s$^{-1}$ between 12 and 22 cm, where wind slabs prevail. Using known values of the water vapor pressures over ice (Marti and Mauersberger, 1993), we computed the fluxes shown in Figure 12. Between 2 and 12 cm, the flux decreases exponentially over time due to snow cooling and the exponential dependence of vapor pressure on temperature. On December 1[st], flux values have decreased to less than a quarter of their November 1[st] values. This is when the snow height increased from 10 to 18 cm (Figure 3) and based on Figure 2 and other pit observations, when deposited snow stopped transforming into depth hoar. The transition from depth hoar to wind slab in Arctic snowpack is almost always very abrupt, so that there is certainly a threshold effect, as already indicated by previous studies (Marbouty, 1980). Given the discontinuous nature of precipitation and snow accumulation (where wind plays a key role), these data suggest that at some point in the season, a snow accumulation episode (whether caused by precipitation or wind) will decrease the temperature gradient and hence the water vapor flux below the threshold triggering the depth hoar to wind slab abrupt transition.

It is interesting to evaluate whether calculated fluxes can explain the mass loss leading to snow collapse. Assuming all the flux comes from a 5 cm-thick depth hoar layer of density 250 kg m$^{-3}$, this represents 12.5 kg m$^{-2}$ of depth hoar. Assuming early season fluxes reached 0.5 mg m$^2$ s$^{-1}$ by extrapolating the data of Figure 12, this leads to a mass loss of 2.6 kg m$^{-2}$ of depth hoar over 2 months, insufficient to explain the near-total disappearance of the depth hoar in some places, so that processes other than purely diffusive fluxes must have been operating. Convection-enhanced fluxes is a possibility, as these may have taken place in the highly permeable depth hoar that formed very early in the season, as evidenced by our thermal conductivity measurements (Figure 3). (Sturm and Johnson, 1991) did observe such convection in depth hoar, and the irregular nature of collapse (Figure 8) is compatible with the presence of convection cells. Another possible factor is wind-induced air advection. Wind-pumping was indeed favored by the rough snow surface, the nearly continuous wind in the early season in both years studied (Figures 3 and 6) and the shallow and highly permeable snowpack. Finally, these estimates are based on the $D_v$ values measured by (Calonne et al., 2011), who, based on details given, do not seem to have studied large-grain depth hoar as found in the Arctic. It cannot be ruled out that vapor diffusion enhancement by the sublimation-condensation cycles (Sturm and Benson, 1997) does not take place in Arctic depth hoar. (Sturm and Benson, 1997) found an average enhancement factor of 4, and if such a factor applied to our case, it would explain the near complete disappearance of the basal depth hoar.

Figure 12 shows that in the upper region, the water vapor flux was much lower and apparently insufficient to allow depth hoar formation. For most of the snow season, there was a continuous upward water vapor flux, leading to overall water vapor loss to the atmosphere. The late season reversal of the flux direction in early May lasted only about a month and was insufficient to reverse the overall loss trend. This reversal coincides with the change in the trend of evolution of $k_{snow}$ at 12 and 22 cm. At 12 cm, $k_{snow}$ started to rise and at 22 cm $k_{snow}$ stopped rising at that moment. We propose that at 12 cm, the water vapor flux led to a density increase and enhanced sintering caused by the growth of bonds between grains under the low temperature gradient conditions (Colbeck, 1998), resulting in a $k_{snow}$ increase. At 22 cm on the other hand, the snow became warmer than the other layers (Figure 3), so that the dominant process switched from condensation to sublimation, leading to a density decrease, halting the rise in $k_{snow}$ or perhaps even producing a slight decrease.

There is also a temperature gradient, and hence a water vapor flux, between the soil and the snow, as already mentioned by (Sturm and Benson, 1997) from observations in interior Alaska. The resulting soil water vapor loss is detectable in Figure 10. In late summer 2013, the liquid water content in the soil at 10 cm depth was about 58%. After thawing in early July 2014, the water content only rises back to 31% meaning that almost half of the water present in the soil the previous summer has been lost by sublimation during the snow season. If this value applies to the top 10 cm of the soil, then it lost 17 kg m$^{-2}$ of water, the same order of magnitude as observed by (Sturm and Benson, 1997), 5 kg m$^{-2}$. Within a few weeks, the water content had risen back to 47% because of precipitation in July 2014. At 2 and 5 cm depth, the loss appears even greater, but the signal is

not as simple, as erratic water percolation due to snowmelt is superimposed onto the thawing signal

## 4.2 Snowpack structure and subnivean life

The presence of a soft depth hoar layer clearly facilitates subnivean travel and food search. The softer the layer, the easier the travel and presumably the better the feeding and reproductive success of subnivean species. Factors that adversely affect the softness of this layer include wind packing and melt-freeze events. Wind storms form hard dense wind slabs. At the base of the snowpack, the strength of the gradient may allow their transformation into indurated depth hoar, but such depth hoar is much harder than that formed in softer snow (Domine et al., 2012). Likewise, early season melt freeze events may lead to ice formation. However, in the Arctic, melt-freeze crusts can transform into depth hoar (Domine et al., 2009). This is probably what happened in autumn 2014, as signs of early season melt-freeze cycling were still observable in May 2015 (Figure 3). Based on these observations, it appears that feeding conditions for subnivean species may have been slightly better in 2013-2014 than in 2014-2015. However, at this stage the evidence is tenuous, and mostly based on spring observations. Thermal conductivity monitoring does not invalidate nor support this conclusion: no NP was placed low enough the first year, and the second year shows a very low value at 2 cm height, except for the first month of snow. It is likely that the higher early season values were due to the presence of a melt-freeze crust that rapidly transformed into depth hoar while the $k_{snow}$ value decreased.

The recent work of (Fauteux et al., in press) however appears consistent with our observations. These authors measured lemming abundances very close to our snow study site right after snow melt in the natural environment and in a very large 9 ha exclosure to minimize predators impact on populations. The exclosure data is therefore expected to be more likely to be affected mostly by just snow conditions. Their data show counts of 6 lemmings ha[-1] in June 2014 vs. 2 lemmings ha[-1] in June 2015, in agreement with our field observations. This correlation is of course very preliminary and only serves to illustrate the interest of monitoring snow conditions to understand lemming population dynamics.

## 4.3 Measurements vs. simulations of snow $k_{snow}$

Our time series of $k_{snow}$ allow us to test the ability of a detailed snow physics model to predict the value of this variable. This test is important, as the water vapor flux is an important process in the shaping of Arctic snowpacks. Giving that detailed snow physics models such as Crocus (Vionnet et al., 2012) or SNOWPACK (Fierz and Lehning, 2001) do not take into account these fluxes, their ability to predict $k_{snow}$ needs testing. A paper on SNOWPACK (Bartelt and Lehning, 2002) states that that model does include water vapor fluxes, but the scheme described was never actually implemented in the model (C. Fierz, private communication, 2015). Figure 13 compares our measurements of $k_{snow}$ with those simulated by Crocus. The Crocus runs performed are those detailed in (Domine et al., 2016) for herb tundra conditions.

It is clear that simulations and measurements yield very different results. Measurements show low values for the lower depth hoar layers and high values for the upper wind slabs. On the contrary, simulations show a value around 0.26 W m[-1] K[-1] for the very basal layer, indicating a melt-freeze crust, values around 0.12 W m[-1] K[-1] for the lower depth hoar layers and values always lower than 0.07 W m[-1] K[-1] for the upper layers. On 13 May, high simulated values appear around 22 cm.

Our proposed interpretation of these differences, aided by a detailed analysis of Crocus output data, is as follows. Crocus simulated a melting episode in late September, giving the basal layer a high thermal conductivity. This is consistent with our observations of a melt-freeze relic in the basal depth hoar layer. However, Crocus cannot predict the transformation of a melt-freeze layer into depth hoar because it does not simulate the required vapor fluxes. These fluxes lead to mass loss in the lower layers and mass gain in the upper ones. This is an important process that contributes to the observed inverted density profiles (Figure 3) (Sturm and Benson, 1997) and hence the inverted thermal conductivity profiles because thermal conductivity is calculated from density only. The other processes that lead to dense upper layers are wind packing and to a

smaller extent weight compaction. The Crocus representation of the wind-packing process cannot be evaluated here, as the density increase also has contributions from water vapor deposition due to the upward flux and their respective contributions cannot be observed separately. An appropriate description of the water vapor flux is required to test the representation of wind packing here. In any case, it is clear that omitting vapor fluxes in Arctic snowpacks leads to an inadequate simulation of the thermal conductivity profile of the snowpack.

A detailed evaluation of the ability of Crocus to reproduce the ground thermal regime is in order. However, ground temperature also depends on soil properties so that coupling to a land surface scheme is required for full testing. Crocus is currently coupled to the land surface scheme ISBA through the SURFEX interface. Improved snow and soil schemes for ISBA are being tested (Decharme et al., 2016) and the evaluation of these new schemes will be the subject of future work.

Lastly, the parameterization of (Yen, 1981) to calculate thermal conductivity from density may not be suitable for Arctic snow. For a density representative of the depth hoar studied, 200 kg m$^{-3}$, (Yen, 1981) predicts $k_{snow}$ =0.11 W m$^{-1}$ K$^{-1}$, while we consistently measured values around 0.025 W m$^{-1}$ K$^{-1}$. The more recent parameterization of (Calonne et al., 2011) predict 0.10 W m$^{-1}$ K$^{-1}$. Both parameterization are based on a small number of values (30 for (Calonne et al., 2011)) obtained on Alpine or temperate snows. By contrast, using the parameterization of (Sturm et al., 1997) based on over 500 values of Arctic or subarctic snows predicts $k_{snow}$ =0.065 W m$^{-1}$ K$^{-1}$. The extensive data set of (Sturm et al., 1997) also shows that for a given density value, the range of $k_{snow}$ values varies by a factor of 4 to 5, and our measured values are within this range. These correlation-based estimated of $k_{snow}$ show that (i) density-thermal conductivity correlations cannot accurately predict $k_{snow}$; and (ii) using parameterizations based on a small data sets consisting mostly of Alpine or temperature snow cannot be used for Arctic snow.

### 4.4 Seasonal variations of $k_{soil}$

Understanding and predicting the seasonal variations of $k_{soil}$ is essential for modelling the thermal regime of permafrost. An important issue is whether $k_{soil}$ can be modelled using a bimodal distribution, i.e. a single value for the thawed state and a single value for the frozen state, or more time-variable values have to be used. (Goodrich, 1986) used an NP fairly similar to ours to measure the seasonal variations in $k_{soil}$ in Canada and concluded that using a bimodal distribution could lead to large errors. (Overduin et al., 2006) monitored $k_{soil}$ in Alaska and observed long transition periods during freezing and thawing. In autumn, the rise between the thawed and frozen values took over a month, while in the spring, complex processes were observed, in particular a spike leading to very high values at the end of the thawing period.

Here, in contrast, we find essentially a bimodal distribution, with transition periods of less than ten days. The transition from thawed to frozen $k_{soil}$ values takes six days in 2013 and eight days in 2014 (remember that measurements are made every two days only). The transition from frozen to thawed $k_{soil}$ values takes 2 days in 2015. In 2014, the data are not of sufficient quality to determine the transition duration accurately. The fast thawed to frozen transition cannot be explained by rapid freezing, as this takes over a month for both years studied. The transition takes place at the end of freezing, and we suggest that the $k_{soil}$ transition can only take place when the water bridges (Sakaguchi et al., 2007) at the soil grain contacts freeze. A significant fraction of conductive heat transfer takes place at grain contacts where water accumulates because of surface tension effects (Penner et al., 1975). These contacts are expected to freeze last because of the small thickness of the water film. Since ice has a much greater thermal conductivity than water, freezing of the contacts is expected to lead to an increase in $k_{soil}$. This hypothesis is comforted by the observation that the $k_{soil}$ transition during thawing takes place at the onset of thawing, and the ice bridges at the grain contacts are here expected to thaw first.

The bimodal distribution observed here contrasts with the studies of (Goodrich, 1986) and (Overduin et al., 2006). It is likely that the duration of the transition depends on soil properties and in particular on grain size distribution and water content. Another factor that may come into play is the heating power used to measure $k_{soil}$. We suggest that it may lead to an artefact if it is too high. We used a power of 0.4 W m$^{-1}$. Neither (Goodrich, 1986) nor (Overduin et al., 2006) specify their heating

power, but most researchers measuring $k_{soil}$ with heated line sources use much higher (and arguably more suitable for obtaining quality data) powers. For example, (Sakaguchi et al., 2007) used 2.7 W m$^{-1}$ to study sandy soils, about 7 times as large as what we used. When we measure $k_{snow}$ in snow close to 0°C, melting can occur and this manifests itself by reduced heating, as most of the heating power is absorbed by latent rather than sensible heat. This translates into an apparent very high (and unreasonable) $k_{snow}$ value, because $k_{snow}$ is inversely proportional to the logarithm of the rate of heating (Morin et al., 2010). The risk of melting is of course reduced is a low heating power is used. In soils, there is a range of melting temperatures but we propose that the fraction of latent vs. sensible heat is reduced if the heating power is low. Moreover, just before the onset of thawing, a high power may induce melting, leading to an apparent $k_{soil}$ value higher than the frozen value, as observed by (Overduin et al., 2006). Melting may also lead to water redistribution, and hence structural changes that may durably modify $k_{soil}$ values in the vicinity of the line source. By limiting the frequency of measurements to once every 48 h and by also reducing the power, we minimize such structural changes and this maximizes the chances that our measured values are representative of those of the studied medium. Of course, we realize that this has a price: heating being reduced, the quality of the data is not as good as when higher powers are used. In any case, we cannot claim based on just these measurements that $k_{soil}$ values can be modelled in general with a bimodal distribution. We merely provide evidence that in our case, we did observe such a distribution. In our opinion, more measurements on different soil types using different heating powers are needed to determine whether (1) the shape of the distribution depends mostly on soil type and properties or (2) non-bimodal distributions are artefacts caused by high heating powers.

**5 Conclusion**

We feel that the following points are important conclusions of this study:

1.  Vertical water vapor fluxes induce by the temperature gradients in the soil and the snowpack strongly determine snow conditions, soil dehydration and the water budget of the surface.

2.  Water vapor fluxes also determine the snow thermal conductivity profile and the ground thermal regime. The comparison of observed vs. simulated thermal conductivity profile demonstrates that omitting these fluxes lead to a radically different snow thermal conductivity profile.

3.  Major snow models (Crocus, SNOWPACK) do not describe water vapor fluxes. The consequences on the water budget, on the ground thermal regime, on the energy budget of the surface, and possibly on climate may be quite significant.

4.  For both years studied, a layer of soft depth hoar was present at the base of the snowpack, which seems to be favorable conditions for subnivean life. In the second year, a melt-freeze layer at the very base of the snow pack may have rendered conditions somewhat less favorable for a few weeks, but $k_{snow}$ monitoring indicates that it transformed rapidly into depth hoar. In May 2015, however, we observed that that basal depth hoar was harder than in May 2014. We note with interest that lemming populations were also higher in spring 2014 than in spring 2015 (Fauteux et al., in press).

5.  According to our data, soil thermal conductivity can be modelled using one value for the thawed period and one value for the frozen period. This is in contrast with previous studies. This may be due to different soil type and properties in different studies. Alternatively, the transition periods between thawed and frozen values may be artificially lengthened when too high heating powers are used for thermal measurements.

**Supplement**

Data used in Figures 3, 5, 6 and 10 are presented in a spreadsheet as a supplement to this paper.

**Author contributions**

F. Domine designed research. D. Sarrazin built and deployed the instruments with assistance from F. Domine. M. Barrere and F. Domine performed the field measurements. F. Domine and M. Barrere analyzed the field data. M. Barrere performed the model simulations. F. Domine prepared the manuscript with comments from M. Barrere and D. Sarrazin.

**Acknowledgements**

This work was supported by the French Polar Institute (IPEV) through grant 1042 to FD and by NSERC through the discovery grant program. We thank Laurent Arnaud for advice in writing the program to run the needle probes. The Polar Continental Shelf Program (PCSP) efficiently provided logistical support for the research at Bylot Island. We are grateful to Gilles Gauthier and Marie-Christine Cadieux for their decades-long efforts to build and maintain the research base of the Centre d'Etudes Nordiques at Bylot Island. Winter field trips were shared with the group of Dominique Berteaux, who helped make this research much more efficient and fun. Bylot Island is located within Sirmilik National Park, and we thank Parks Canada and the Pond Inlet community (Mittimatalik) for permission to work there. The assistance of Matthieu Lafaysse, Samuel Morin and Vincent Vionnet at Centre d'Etudes de la Neige (Météo France-CNRS) for the use of Crocus is gratefully acknowledged. Constructive reviews by Matthew Sturm, Martin Schneebeli and an anonymous reviewer are gratefully acknowledged.

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

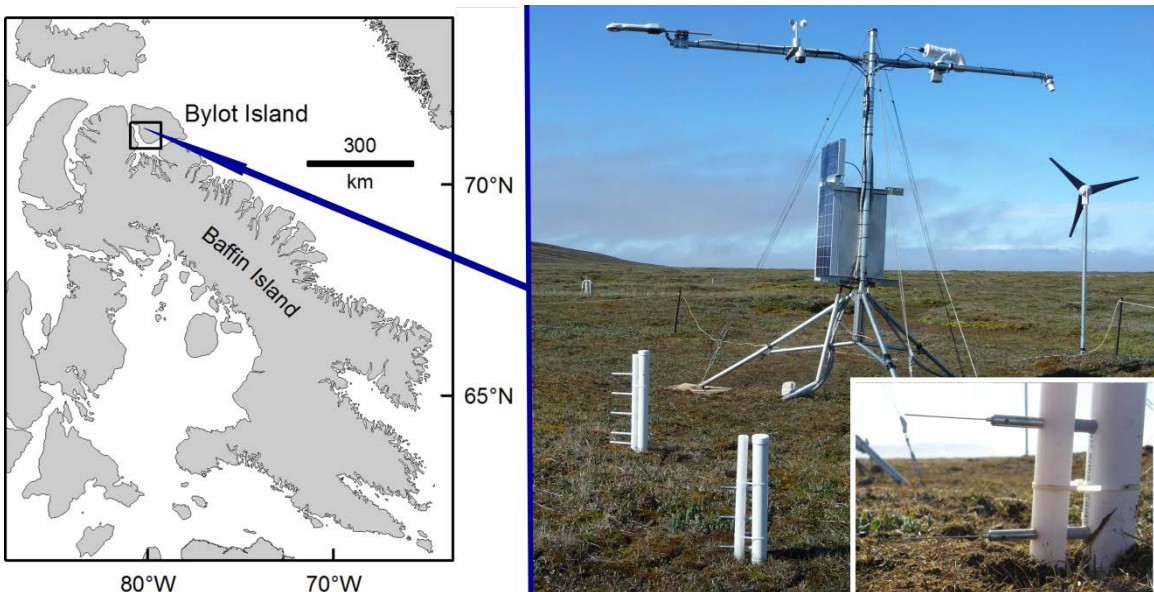

**Figure 1.** Location of the study site on Bylot Island in the Canadian Arctic archipelago and photograph of the monitoring station deployed. The polyethylene post with the three TP08 heated needle probes is in the foreground. The polyethylene post with the five thermistors is visible behind it. The radiometers, SR50 snow height gauge, cup anemometer, temperature and relative humidity gauge, and surface temperature sensors are visible on the tripod, from left to right. The CR1000 data loggers and batteries are in the metal box on the tripod. Batteries are recharged by solar panels and a wind mill in winter. Inset: detail of the lower two TP08 needles after their positions were lowered in July 2014.

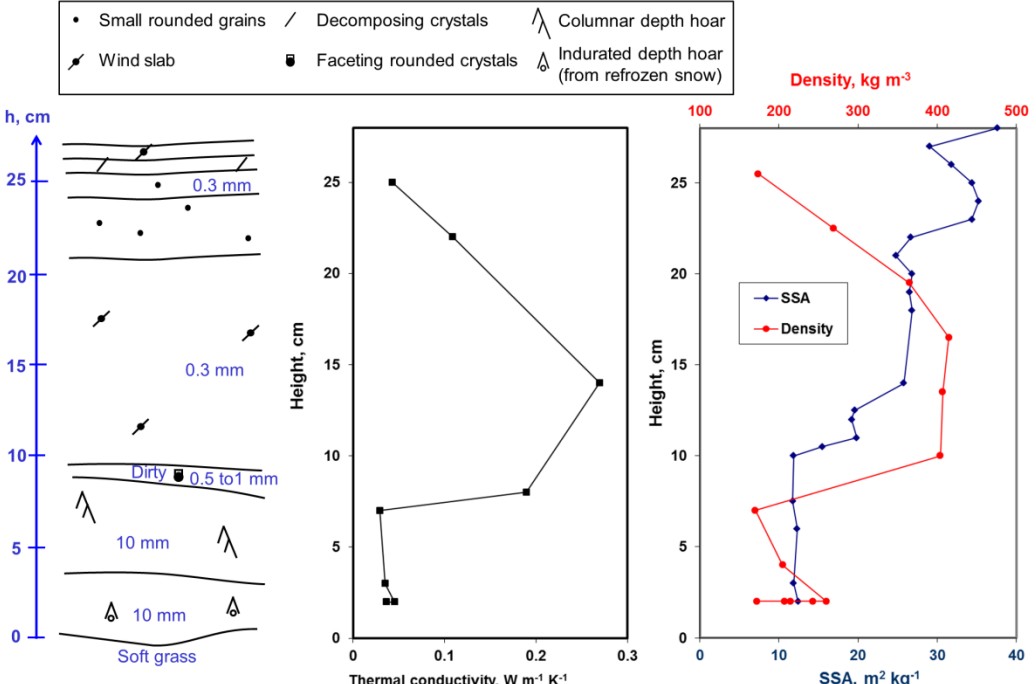

**Figure 2.** Stratigraphy and vertical profiles of snow physical properties near our study site on 12 May 2015. Visual grain sizes are indicated in the stratigraphy. Density data are for the middle of the 3 cm-high sample. Snow type symbols are those of (Fierz et al., 2009), except for the basal melt-freeze layer which transformed into depth hoar to form an indurated layer, as detailed in the text.

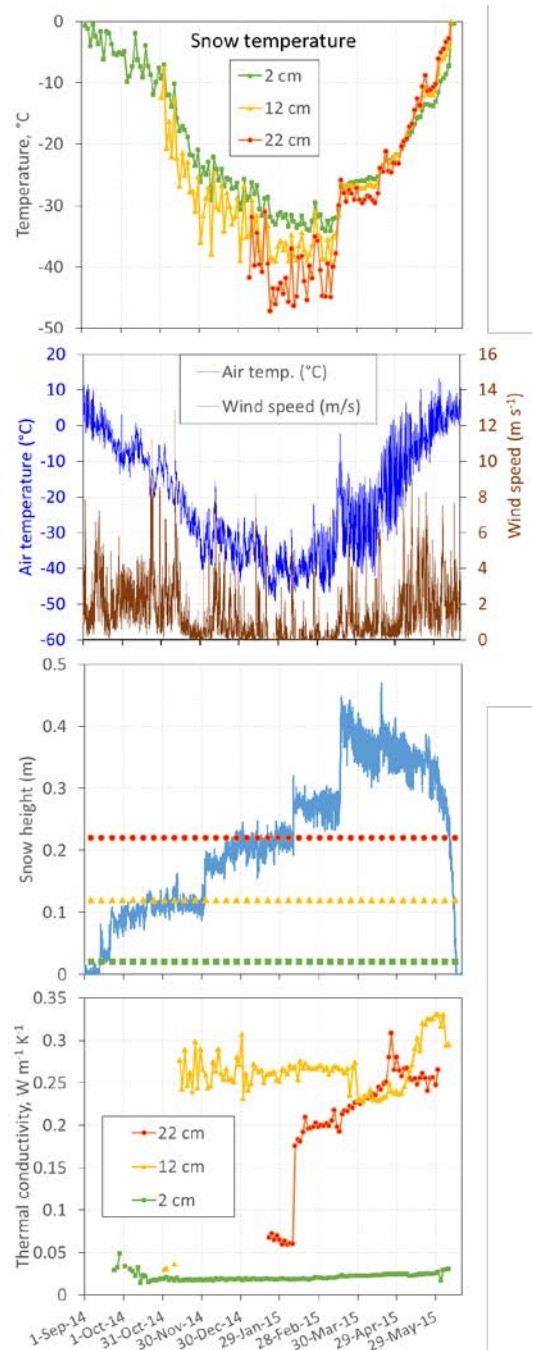

**Figure 3.** Snow temperature, air temperature, wind speed, snow height and snow thermal conductivity at three heights during the 2014-2015 winter season at Bylot Island. The levels of the three thermal conductivity needle probes (NPs) are indicated in the snow height panel. Note that the snow height gauge and the NPs were about 5 m away, so that snow height between both spots may have been different. Snow temperatures were measured with the NPs every other day at 5:00.

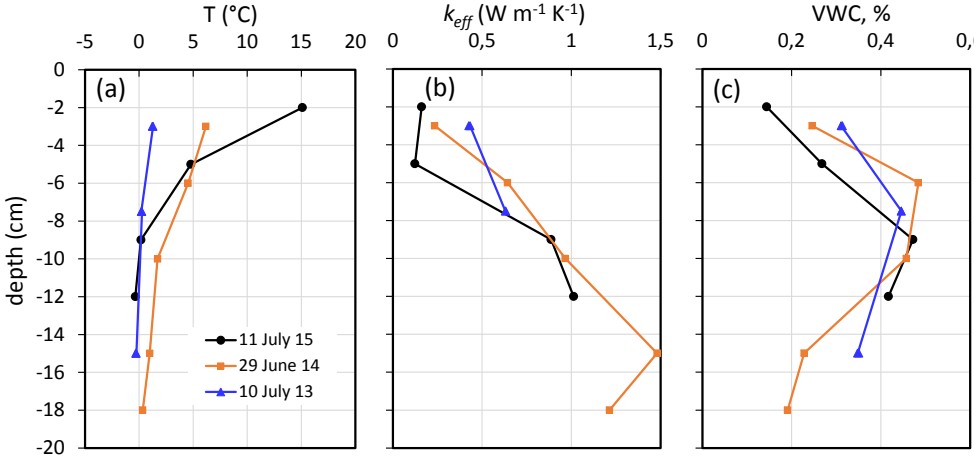

**Figure 4.** Vertical profiles of soil physical variables in pits dug in the polygon where our instrument station is located, during summers 2013 to 2015. (a) Temperature; (b) thermal conductivity; (c) volumetric liquid water content.

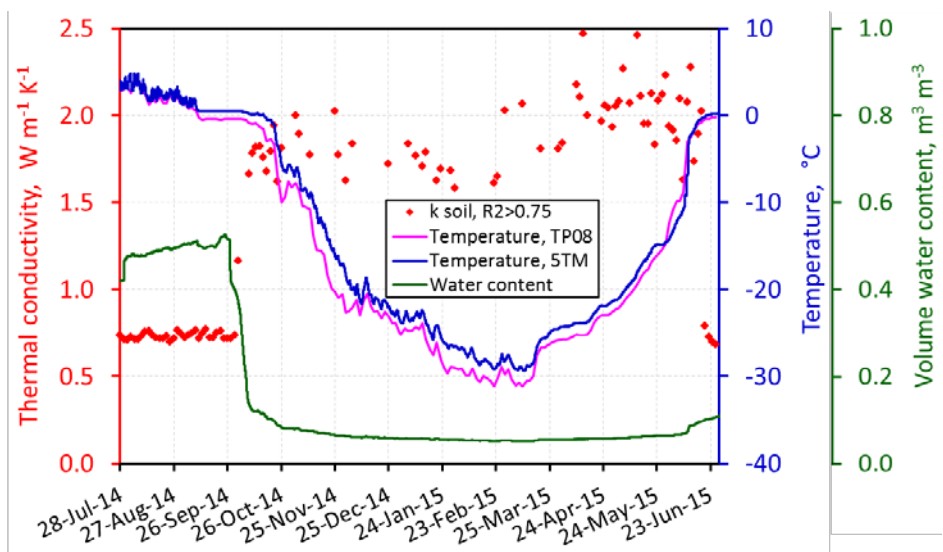

**Figure 5.** Seasonal evolution of the thermal conductivity, temperature and volume water content of the soil at 10 cm depth for the 2014-2015 season. The 5TM probes which measures both temperature and water content hourly is about 2 m from the TP08 NP, which measures thermal conductivity and temperature every 2 days.

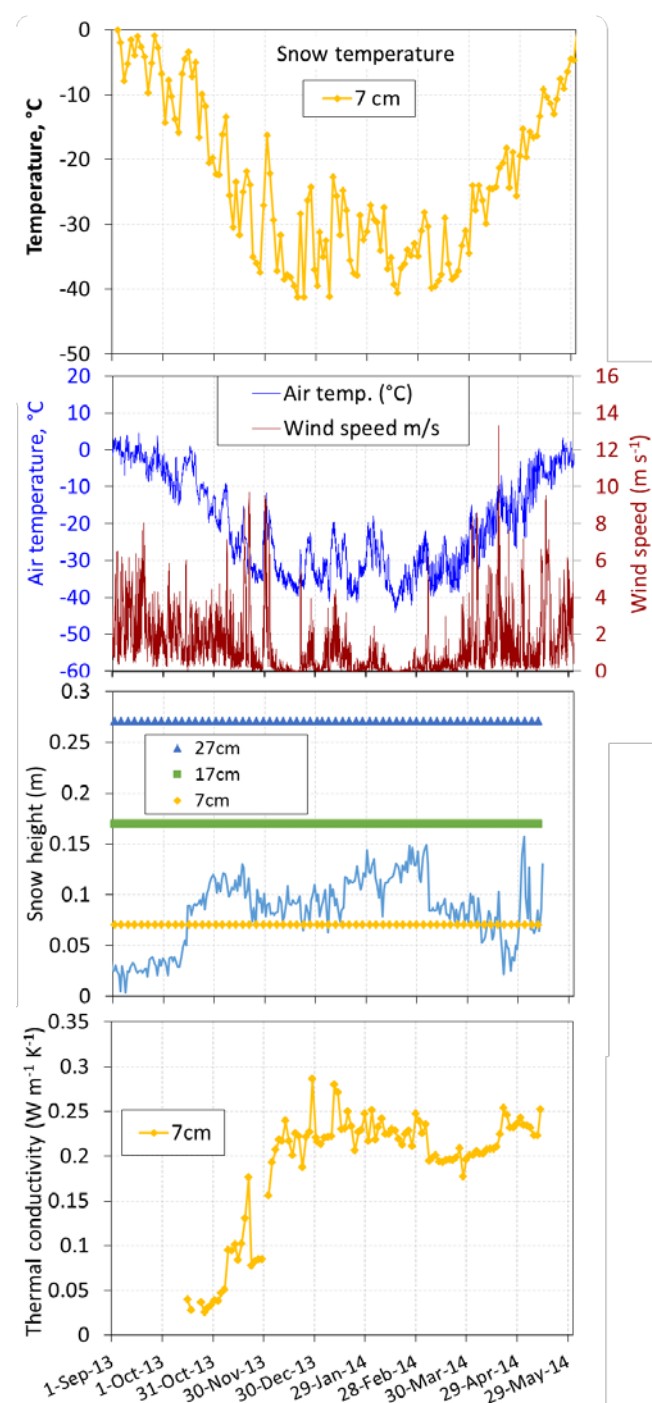

**Figure 6.** Snow temperature, air temperature, wind speed, snow height and snow thermal conductivity at 7 cm height during the 2013-2014 winter season at Bylot Island. The levels of the three thermal conductivity needle probes (NPs) are indicated in the snow height panel, showing that only the lowermost NP was covered.

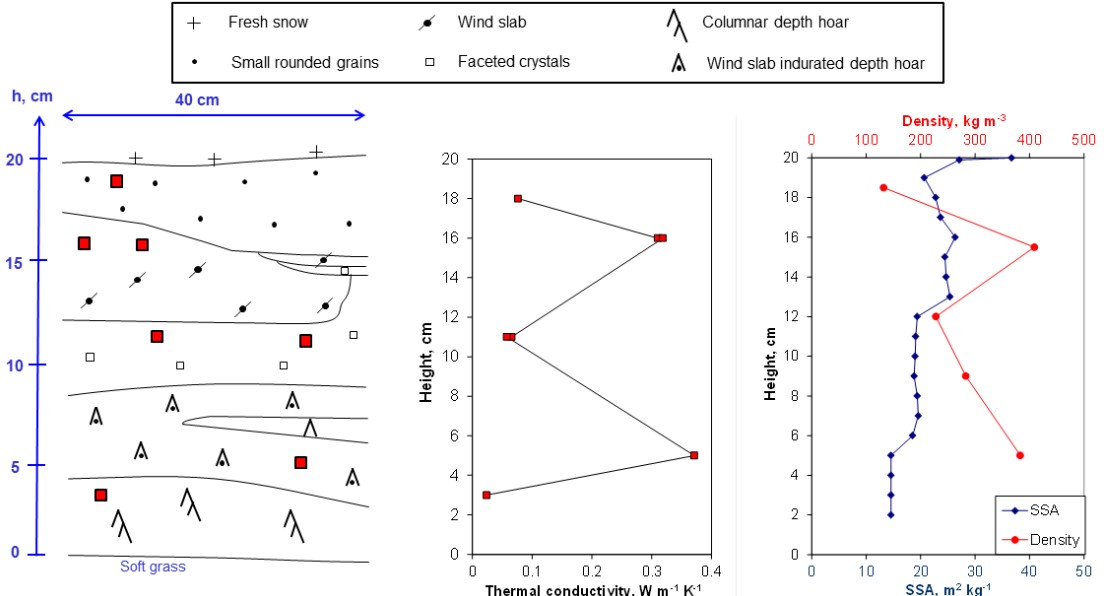

**Figure 7.** Stratigraphy and vertical profiles of snow physical properties near our study site on 14 May 2014. Fresh snow is just a 2 mm-thick sprinkling, also visible in the SSA profile. Density data are for the middle of the 3 cm-high sample. Snow type symbols are those of (Fierz et al., 2009), except for the lower wind slab which transformed into depth hoar to form an indurated layer, as detailed in the text. Red-filled black squares in the stratigraphy indicate where thermal conductivity measurements were made.

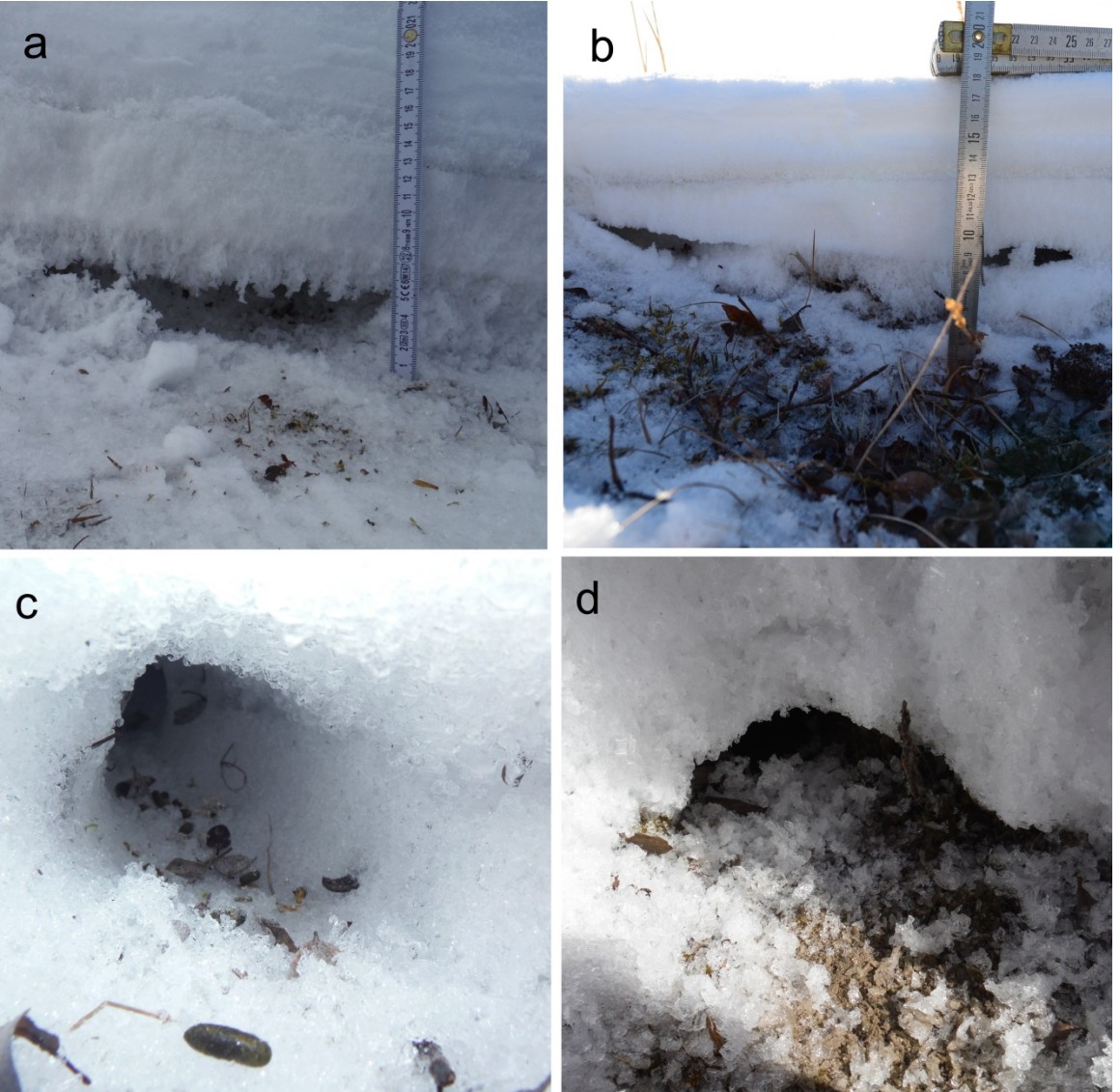

**Figure 8**. Gaps in the snow in the basal depth hoar layer. (a) and (b) gaps due to the spontaneous collapse of the depth hoar, following season-long mass loss because of the upward water vapor flux. (c) and (d) lemming burrows, easily identifiable by their regular shape and the presence of characteristic feces (c, foreground).

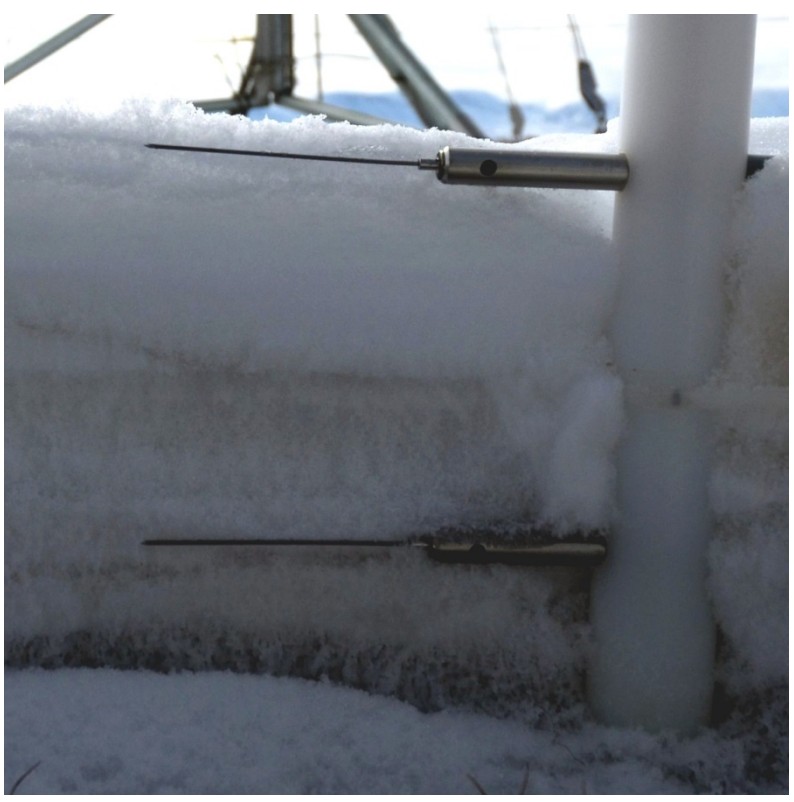

**Figure 9.** Photograph of the snow stratigraphy taken on 14 May 2014. The NPs are 7 and 17 cm above the ground. The various depth hoar and indurated depth hoar layers between 0 and about 11 cm are clearly visible, as well as the wind slab between 11 and 16 cm.

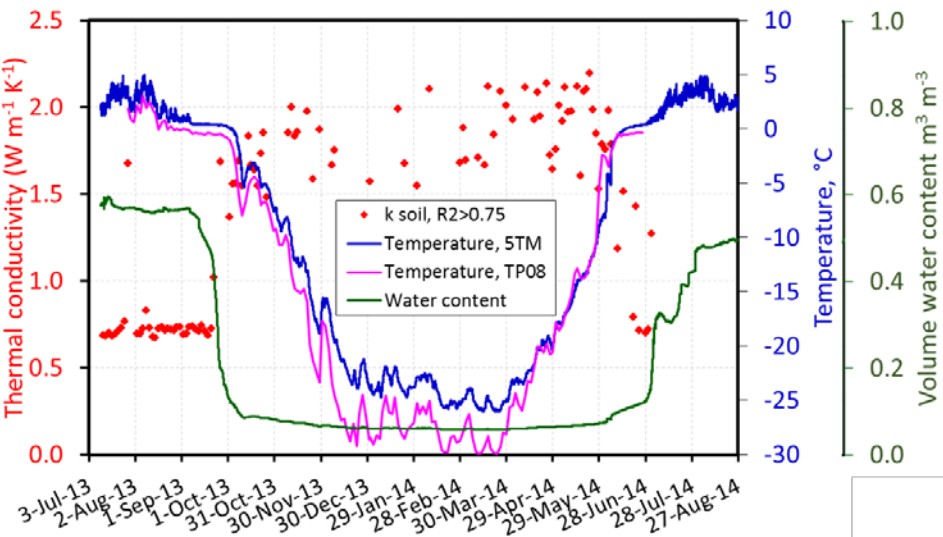

**Figure 10.** Seasonal evolution of the thermal conductivity, temperature and volume water content of the soil at 10 cm depth for the 2013-2014 season. The 5TM probes which measures both temperature and water content hourly is about 2 m from the TP08 NP, which measures thermal conductivity and temperature every 2 days.

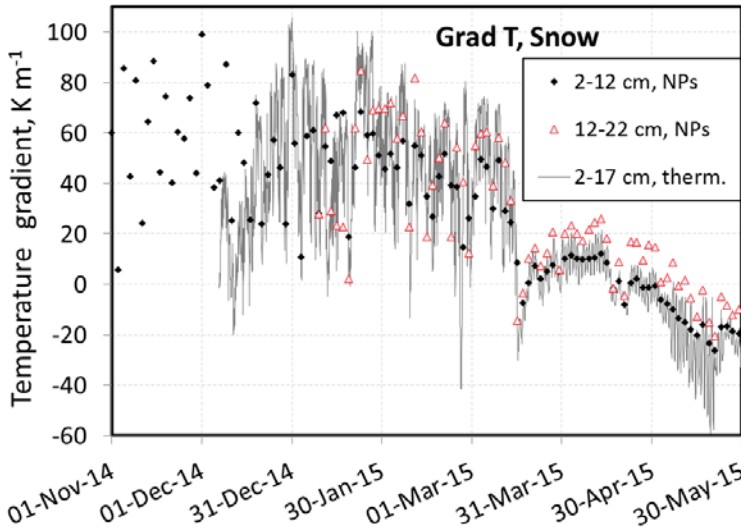

**Figure 11.** Time series of the temperature gradient in the snow. Values were obtained from the heated needle probes at 2, 12 and 22 cm, with a data point every 2 days. Thermistors at 2 and 17 cm also measured temperature every hour, and the values are shown with an hourly resolution. The different start dates of each curve are determined by the date where the snow height reached the relevant level.

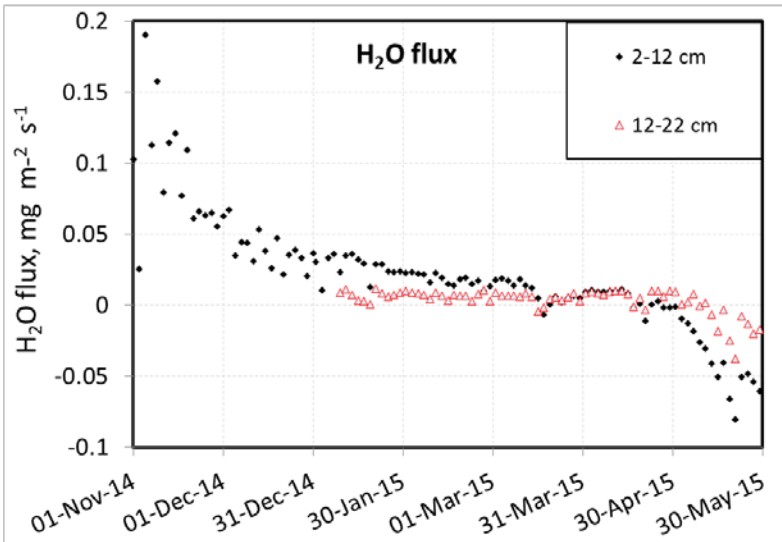

**Figure 12**. Time series of the water vapor flux at two levels in the snowpack. Positive fluxes are upward.

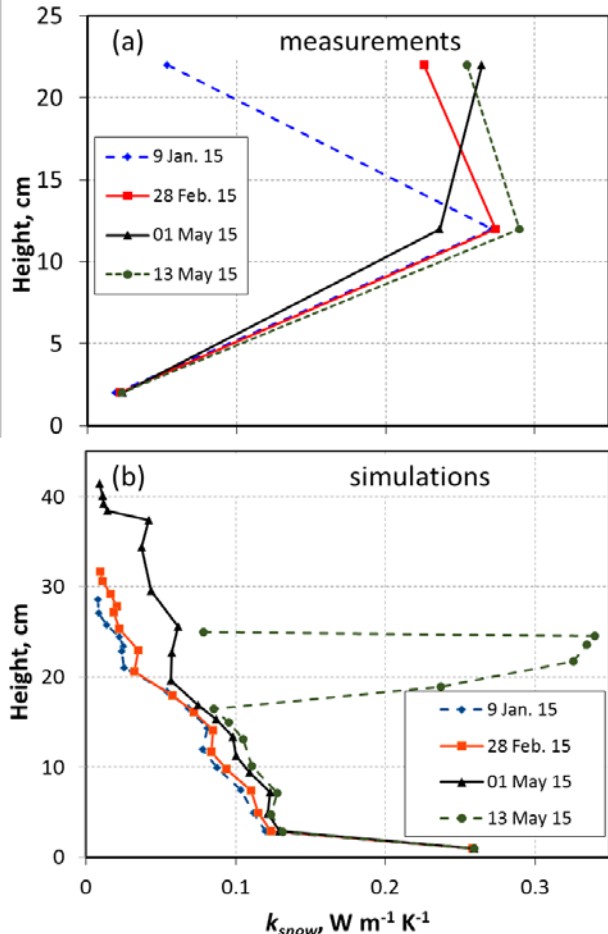

**Figure 13**. Vertical profiles of $k_{snow}$. (a) Measured by the needle probes; (b) modelled by CROCUS.