# Peer review of "Seasonal evolution of the effective thermal conductivity of the snow and the soil in high Arctic herb tundra at Bylot Island, Canada"

_The Cryosphere, 2016_

## Referee Comment (RC1) · Anonymous Referee #1 · 2 Jun 2016

Seasonal evolution of the effective thermal conductivity of the snow and the soil in high Arctic herb tundra at Bylot Island, Canada

F. Domine, M. Barrere and D. Sarrazin

General comments:

The paper analyzes measurements of thermal conductivity of snow and soil in the Arctic, specifically at Bylot Island, Canada. Measurements occurred through 3 field seasons, and values compared to other studies, and to modeled values from the French snow model Crocus. Results are very interesting since it is suggested that a bi-modal distribution can be used to characterize the soil thermal conductivity. Simulated values

from Crocus are contrary to measurements, the authors explain this by the absence of upward vapor flux from temperature gradient in the model.

Overall, the analysis and measurements are thorough and are quite relevant for the improvement of snow modeling in the Arctic, which remains poor. This paper provides insights on improvements needed for a better assessment of climate variability in the Arctic. I recommend this paper for publication in The Cryosphere, after minor revisions detailed below.

Specific comments:

Section 2.1: The authors describe the vertical locations of NPs and thermistors. One was moved following observations of depth hoar height. Furthermore, the chosen height for the other NPs and thermistors seem to be motivated by stratigraphy, which makes sense. However, such detail is not mention. It would be nice if the authors explain the reasoning behind the 2-12-22cm heights for NPs, and the 2-7-17-37 for thermistors? Also, why are thermistors at different height than NPs? They seem aligned with the initial positions of the NPs (7-17-27), why were they not lowered to match the NPs?

I will leave to the discretion of the author to include a map of Bylot to Figure-1.

Line 142: I would clarify why depth hoar are more conductive in the vertical direction (i.e. owing to the higher thermal conductivity of ice relative to air, and an elongated grain). . . And why is it more conductive horizontally in wind slab?

Section 2.3: Although the modeling is described in details elsewhere, I would put a little more details here, otherwise the section can be removed and stated elsewhere since it is only 2-3 sentences long. The authors could simply add the data that was missing, the reasoning using ERA, and how Crocus computed thermal conductivity. The conclusions are significant with regards to Crocus, and more context need to be provided here for the reader's understanding.

Figure-2 has very poor resolution and consequently very hard to read.

Section 3.1: I assume SSA are from DUFISSS? This should be mentioned in the paper... Figure-6: dates are in French

Section 4.3.: A figure on Crocus output, visualized profile with marks on the melt event would greatly help the understanding of this section. The authors could display Crocus thermal conductivity and snow temperature (or temperature gradient). The problem of density profiles and simulated grains would be more obvious...

---

## Referee Comment (RC2) · M. Sturm (Referee) · 14 Jun 2016

Review of *Seasonal evolution of the effective thermal conductivity of the snow and the soil in high Arctic herb tundra at Bylot Island, Canada*

Journal: The Cryosphere (TC) Title: Seasonal evolution of the effective thermal conductivity of the snow and the soil in high Arctic herb tundra at Bylot Island, Canada Author(s): F. Domine, M. Barrere, and D. Sarrazin MS No.: tc-2016-107 MS Type: Research article

In this paper the authors report on the evolution of the thermal conductivity ($k_{eff}$) of the tundra snow cover of Bylot Island. It is an interesting paper, with a key finding being that the contrast in $k_{eff}$ between depth hoar and wind slab is on the order of 1:10. While this finding is not new, it is a useful piece of information that is neither widely appreciated nor captured by snow models (as the authors show). Overall, I recommend the paper be published….but not until it has been shortened considerably. The number of readers who will want to wade through all the detail currently in the paper in order to glean the main points is limited, and as consequence, the impact of the paper will be reduced

One other key point: if the 1:10 contrast in $k_{eff}$ is the big take-away message of the paper, then the authors have missed an opportunity to discuss a key aspect of how tundra snow functions, in a way that is convolved with this 1:10 contrast. As seen in the authors' Figures 2 and 7, the relative percentage of depth hoar vs. wind slab controls the bulk thermal conductivity of tundra snow covers to a large extent. The number of windstorms, the relief of the underlying tundra tussocks (inter-tussock depth hoar), and the sequence of snowfall events all combine to determine what these percentages will be, hence how conductive the snow cover will be. This is a sensitive balance, one that could easily change if the climate, wind regimes, and snow-up dates, change. Discussing this would be valuable and for those readers that this paper aims at possible new information.

Other specific comments follow:

1. The authors have a tendency to cite new rather than seminal work, and philosophically, I find this distressing. It leads to the field "forgetting" facts and findings, then these have to be rediscovered. For example, that there are large amounts of organic matter in the permafrost was well known and reported on in the 1960s and 1970s: see Lachenbruch and Ferrians among other authors. A similar comment on references in the snow literature.

2. Line 117: Convection in snow. This was a lot of attention paid this topic in the 1980s and 1990s. The authors should see and cite:
   - Powers D, O'Neill K, Colbeck SC (1985) Theory of natural convection in snow. Journal of Geophysical Research 90:10641-10649.
   - Sturm M, Johnson JB (1991) Natural convection in the subarctic snow cover. Journal of Geophysical Research 96 (B7):11657-11671.
   - Sturm M (1991) The Role of Thermal Convection in Heat and Mass Transport in the Subarctic Snow Cover. USA-CRREL Report 91-19.

3. Line 151: Alternate methods of measuring thermal conductivity: It is not clear that guarded hotplate methods would produce more accurate or appropriate values $k_{eff}$ ---just different. While the dynamic method of the needle probe has some problems (pointed out by the authors) so too do steady-state methods. For example, real snow covers rarely, if ever, develop a steady-state non-varying temperature gradient. Also, when snow is subjected to uni-directional gradients, the thermal conductivity evolves through

metamorphism, leading to a varying value of $k_{eff}$ (or alternately, such mild gradients have to be imposed that the resultant tests are equivocal). I suggest altering the statement to indicate that the alternate methods are impractical, and *may not even be more accurate*.

4. Line 156: Do you have to worry about pore water migration in measuring thermal conductivity of the soil using a needle probe?

5. Lines 185 and 265-270: ***Indurated depth hoar***. I would like to see a more comprehensive discussion of this snow texture. I believe the lead author and I discussed this snow texture when working jointly on snowpits in Alaska about a decade ago. He is correct that no formal symbol for the material exists in the International Classification, but that arises in part from the fact that I don't think the committee charged with revising the classification believed such a material existed. The symbol I have been using for almost 20 years for this type of snow is a combined symbol of wind slab (black circle with a slash) and depth hoar (chains of grains or just cups). This is not that different than the one introduced in the paper, and it would be good to mention both as many of my students continue to use our older symbol. In addition, Carl Benson and I described this texture in our 1993 paper the phenomenon, though it was a few years later I introduced the term "indurated":

> *Elsewhere wind slabs are adjacent, one on top of the other. As the winter progresses even dense wind slabs can begin to metamorphose into depth hoar. We have observed wind slabs as dense as 0.35 g cm metamorphose into depth hoar by the end of the season.*

Before introducing that term I corresponded with Dr. Akitaya (arguably at the time *the* expert on depth hoar) regarding it. He had never seen this texture in Japan because the temperature gradients are too low to produce it.

6. **Temperature Gradients**: (Lines 313-317) In order to turn wind slabs into indurated depth hoar, strong gradients are needed (see authors Figure 11). While the gradients a Bylot Island are super-critical (> 25°C/m) for much of the winter, they are less strong than I would have expected. I attach comparable data from Alaska's North Slope. Not only are the gradients stronger than those in the paper, but they start earlier in the winter. In a second graph, I have smoothed the data (48 hour running average), as the authors have done. The result is noticeably milder. I suggest the following changes/additions: a) don't smooth the gradients. In 48 hours a lot of metamorphism can take place. Smoothing makes the graph cleaner but less "physical" as far as metamorphism is concerned, b) show the critical gradient….whether that of Marbouty or Colbeck or Armstrong. It gives readers an appreciation of what drives the changes, c) consider computing something like an integrated metric called "time under a strong gradient" or the like. Right now the paper mainly focuses on the agreement between the thermistor and NP values, but that misses the BIG point of what is actually driving the physics. And while Lines 325-330 provide the vapor gradient (and state that it can grow depth hoar), the values are really meaningless unless they are translated into something like a growth rate. Consider as an example computing how fast these vapor gradients could grow a thin sheet of ice…like the skirt on depth hoar cup. That would make the values mean something.

[Figure]

[Figure]

8. **Soil to Snow Fluxes**: These have been measured during depth hoar metamorphism. See:
   - Trabant D, Benson CS (1972) Field experiments on the development of depth hoar. *Geological Society of America Memoir 135:*:309-322.
   - Santeford HS Snow Soil Interactions in Interior Alaska. in Modeling of Snow Cover Runoff, USACRREL, Hanover, NH, pp. 311-318.
   - Sturm M, Benson CS (1997) Vapor transport, grain growth and depth hoar development in the subarctic snow. Journal of Glaciology 43:42-59.

9. Line 341: There is a large literature on depth hoar and subnivean animals. Possibly the best starting point is W. O. Pruitt's 1957 paper:
   - Pruitt Jr, W. O. (1957). Observations on the bioclimate of some taiga mammals. *Arctic*, *10*(3), 130-138.

---

## Referee Comment (RC3) · M. Schneebeli (Referee) · 8 Jul 2016

M. Schneebeli (Referee)

schneebeli@slf.ch

General comments

The is an interesting study, which shows in detail the enormous problems to measure, monitor and interpret thermal conductivity of snow under field conditions in the high arctic. The paper shows that large uncertainties exist in measurement and in the application of models, and that a continuous monitoring is difficult. In fact, the results suggest that simple density measurements and the now very well calibrated parameterizations, maybe a more feasible and precise way to observe the evolution of the snowpack. The interpretation of the measured thermal conductivities of the needle probe are in my view not always supported by other the other data presented in the

paper, as will be discussed below in detail.

The discussion on subnivean life is a bit out of focus in this paper, clearly an important aspect of the arctic snow cover, but in my view not the right place.

———————————————— Specific questions:

The authors interpret thermal conductivities in snow around 0.02 W m-1 K-1 as snow conductivities (they mention that this is within errors the same value as in air). I believe there are two points not made clear: The snowpack, if the bottom layer would be air over an extended area, would immediately compact (in fact, avalanche formation mechanics gives an upper bound of about max. 1 mˆ2 air gap before an spontaneous collapse of the snowpack forms). The "close-to-air" values are therefore at least not spatially representative.

The inclusion of the soil in the interpretation is very useful, except that no detailed granulometric soil analysis seems to exists as this is a well investigated research site? More detailed data would clarify the observed behavior of the soil-freezing behavior. In fact, the observed curve indicates that the soil is not a silt, but a fine sand.

The authors put substantial weight on the effect of water vapor fluxes on the snow-cover. The explanation of the fragile depth hoar bottom layer, as well as the formation of indurated layers, is based on the interpretation of temperature and vapor pressure gradients. The calculation of the vapor flux is omitted with the argument that the diffu-sivity is not well known. Laboratory experiments and numerical simulations (Calonne, 2014; Pinzer, 2012) defined the diffusion coefficient precisely - in fact, due to the hand-to-hand process, the diffusivity in air is a very precise approximation. Approximate calculation for the season 2014-2015, with an average snow temperature of -30 deg C, temperature gradient 50 K m-1, and a duration of 90 days, result in a mass flux of 0.24 kg m-2. This flux seems to me too small to explain the observed processes.

Obviously, spatial variability of the thermal conductivity of snow can not be measured by

permanent stations. However, as is obvious from the snow profile and the descriptions, spatial variability is an issue at the dm - m scale. As a suggestion, long snow profiles as done by Rutter et al (2014) in the arctic, or as demonstrated using a penetrometer by Proksch et al., would have contributed much to reduce the uncertainty in the measured values and their interpretation.

The use of needle probes as monitoring devices is strongly defended by the authors. However, a careful inspection of their Fig. 2 and Fig. 13 a) and calculating thermal conductivity based on the well accepted Calonne et al (2011) parameterization (or the Yen-parameterization) using the measured density, shows that the needle probes underestimate severely (for depth hoar a factor of about five) the effective thermal conductivity.

The numerical simulation using Crocus seems to have major problems with creating a realistic density profile. As no details are given, my conclusion is that severe deficiencies must exist in the model parameterization. I suggest that the model runs are checked by an expert, as they seem to me beyond any reasonable behavior, or this part of the manuscript should be deleted.

References Calonne, N., C. Geindreau, and F. Flin (2014), Macroscopic Modeling for Heat and Water Vapor Transfer in Dry Snow by Homogenization, The Journal of Physical Chemistry B, 118(47), 13393–13403, doi:10.1021/jp5052535.

Pinzer, B. R., M. Schneebeli, and T. U. Kaempfer (2012), Vapor flux and recrystallization during dry snow metamorphism under a steady temperature gradient as observed by time-lapse micro-tomography, The Cryosphere, 6(5), 1141–1155, doi:10.5194/tc-6-1141-2012.

Proksch, M., H. Löwe, and M. Schneebeli (2015), Density, specific surface area, and correlation length of snow measured by high-resolution penetrometry, Journal of Geophysical Research: Earth Surface, 120(2), 346–362, doi:10.1002/2014JF003266.

Rutter, N., M. Sandells, C. Derksen, P. Toose, A. Royer, B. Montpetit, A. Langlois, J. Lemmetyinen, and J. Pulliainen (2014), Snow stratigraphic heterogeneity within ground-based passive microwave radiometer footprints: Implications for emission modeling, Journal of Geophysical Research: Earth Surface, 119(3), 550–565, doi:10.1002/2013JF003017.

——————————————— Technical corrections

l 200 These values are questionable based on the authors density measurements. If there is no heat conducting matrix, there is no mechanical (compressive) strength.

l 230 Did the authors any calibration of the temperature and soil humidity sensors before or after the deployment?

l 283 "Rise" -> rise

l 290 The same limitations concerning vapor flux are valid also for convection (if there is any with the measured snow profile). In my view the speculation is out of place.

l 317 ff The fluxes are easy calculate, this section should be rewritten in view of the actual fluxes.

Almost all Figures: The time axis is lettered in French, not English

Fig. 1 The appearance of the vegetation in the photo seems to involve some vertical structure, completely flattened out during early winter? Not unimportant for the interpretation of the depth hoar formation.

Fig. 2 The symbol for melt-freeze indurated depth hoar is actually defined (Int. Class., p. 19, a lying "8" with depth hoar symbols inside)

Fig. 3 Snow depth or snow height. Caption, text and axis are not consistent (also Fig. 6)

Caption Fig 3: where there no easy measurements of snow depths around the stations

to know spatial variability around?

Fig. 7 The measured thermal conductivity data are inconsistent with the density profile. Give error bars.

---

## Author Comment (AC1) · 2 Sep 2016

**Reply to Reviewer 1**

*The Reviewer's comments are in black, and our response is embedded in the text, in blue italics. Line numbers refer to those of the version in track changes mode.*

General comments: The paper analyzes measurements of thermal conductivity of snow and soil in the Arctic, specifically at Bylot Island, Canada. Measurements occurred through 3 field seasons, and values compared to other studies, and to modeled values from the French snow model Crocus. Results are very interesting since it is suggested that a bi-modal distribution can be used to characterize the soil thermal conductivity. Simulated values from Crocus are contrary to measurements, the authors explain this by the absence of upward vapor flux from temperature gradient in the model.

Overall, the analysis and measurements are thorough and are quite relevant for the improvement of snow modeling in the Arctic, which remains poor. This paper provides insights on improvements needed for a better assessment of climate variability in the Arctic. I recommend this paper for publication in The Cryosphere, after minor revisions detailed below.

*Thank you for these positive comments.*

**Specific comments:**

Section 2.1: The authors describe the vertical locations of NPs and thermistors. One was moved following observations of depth hoar height. Furthermore, the chosen height for the other NPs and thermistors seem to be motivated by stratigraphy, which makes sense. However, such detail is not mention. It would be nice if the authors explain the reasoning behind the 2-12-22cm heights for NPs, and the 2-7-17-37 for thermistors? Also, why are thermistors at different height than NPs? They seem aligned with the initial positions of the NPs (7-17-27), why were they not lowered to match the NPs?

*NPs were initially placed arbitrarily before we had a chance to observe the snow. Subsequently, thermistors were not moved to allow heat flux calculations. This is now mentioned lines 84 and 87.*

I will leave to the discretion of the author to include a map of Bylot to Figure-1.

*We have now added a map.*

Line 142: I would clarify why depth hoar are more conductive in the vertical direction (i.e. owing to the higher thermal conductivity of ice relative to air, and an elongated grain). . . And why is it more conductive horizontally in wind slab?

*This is detailed at length in the references given (Calonne et al., 2011; Riche and Schneebeli, 2013), and we feel that repeating this here would unnecessarily lengthen the paper.*

Section 2.3: Although the modeling is described in details elsewhere, I would put a little more details here, otherwise the section can be removed and stated elsewhere since it is only 2-3 sentences long. The authors could simply add the data that was missing, the reasoning using ERA, and how Crocus computed thermal conductivity. The conclusions are significant with regards to Crocus, and more context need to be provided here for the reader's understanding.

*We have added a brief sentence regarding ERA and mention how thermal conductivity is calculated in Crocus, using a simple correlation with density (lines 179-181).*

Figure-2 has very poor resolution and consequently very hard to read.

*We have improved Figure 2.*

Section 3.1: I assume SSA are from DUFISSS? This should be mentioned in the paper .

*Yes, this was already mentioned in the methods section (now line 165)*

Figure-6: dates are in French

*Not just Figure 6. This has been corrected throughout.*

Section 4.3.: A figure on Crocus output, visualized profile with marks on the melt event would greatly help the understanding of this section. The authors could display Crocus thermal conductivity and snow temperature (or temperature gradient). The problem of density profiles and simulated grains would be more obvious.

*Do we really need an extra Figure while other Reviewers request condensation? One of the main points is really that Crocus predicts melting in the early season and does not predict the transformation of the melt-freeze crust into depth hoar, and this is well detailed. Note that if the editors request so, we will add a Figure showing the evolution of snow stratigraphy over the season, as predicted by Crocus, and the reader can compare that to Figure 2. The danger is that since spatial variations are important, over-interpretations of the differences may be made.*

---

## Author Comment (AC2) · 2 Sep 2016

**Reply to Martin Schneebeli**

*The Reviewer's comments are in black, and our response is embedded in the text, in blue italics. Line numbers refer to those of the version in track changes mode.*

**General comments**

The is an interesting study, which shows in detail the enormous problems to measure, monitor and interpret thermal conductivity of snow under field conditions in the high arctic. The paper shows that large uncertainties exist in measurement and in the application of models, and that a continuous monitoring is difficult. In fact, the results suggest that simple density measurements and the now very well calibrated parameterizations, maybe a more feasible and precise way to observe the evolution of the snowpack. The interpretation of the measured thermal conductivities of the needle probe are in my view not always supported by other the other data presented in the paper, as will be discussed below in detail.

*Thank you for this overall positive comment. We however do not agree with the suggestion that density measurement and the use of density-thermal conductivity correlations would be a good method to estimate thermal conductivity. We discuss this below with the relevant comment.*

The discussion on subnivean life is a bit out of focus in this paper, clearly an important aspect of the arctic snow cover, but in my view not the right place.

*This is a matter of view point. The reviewer is an Alpine snow scientist and some of his main interests include avalanches and snow microstructure. We focus on Arctic problems and subnivean life is critical. In fact, although the main motivation of this program is permafrost thermal regime, another important objective is subnivean life. Please remember that lemmings form the very base of the terrestrial food web (Gauthier et al., 2011) and understanding their life conditions is critical for any Arctic wildlife dynamics consideration. Most Arctic snow scientists always have subnivean life in the back of their mind, but on the other hand very few ever think about avalanches. We do intend to use snow thermal conductivity as a proxy for lemming-relevant snow properties, as explained in our paper (lines 63-64) and we therefore feel this discussion fully belongs to this paper. In fact, a very recent study by (Fauteux et al., in press) shows population variations between 2014 and 2015 that appear consistent with our snow observations and we have added a short paragraph in the discussion to mention this (lines 433-438).*

—————————————— **Specific questions:**

The authors interpret thermal conductivities in snow around 0.02 W m$^{-1}$ K$^{-1}$ as snow conductivities (they mention that this is within errors the same value as in air). I believe there are two points not made clear: The snowpack, if the bottom layer would be air over an extended area, would immediately compact (in fact, avalanche formation mechanics gives an upper bound of

about max. 1 m^2 air gap before an spontaneous collapse of the snowpack forms). The "close-to-air" values are therefore at least not spatially representative.

*The reviewer probably misunderstood our statements. We say clearly in the results section that these values are for depth hoar and that they are a bit low and close to that of air because of the negative artefact due to the use of the NP method. Our lines 227-229 read: "$k_{snow}$ dropped to values around 0.02 W m$^{-1}$ K$^1$ because of rapid depth hoar formation. These values may seem a bit low, especially considering that air has a thermal conductivity of 0.023 W m$^{-1}$ K$^1$, but the low value can be attributed to a negative systematic error of about 20% caused by the NP method, as described in (Riche and Schneebeli, 2013) and discussed above."*

*Regarding the presence of air at the bottom of the snowpack, the reviewer probably misunderstood our statements. We never say that air is continuously present over extended areas. Rather we clearly show the discontinuous nature of these air gaps in Figure 8. Furthermore, spontaneous collapse of the snowpack is predicted for Alpine conditions where snow is thick and the overburden significant. In the Arctic, very different conditions prevail with a thin snowpack (as clearly stressed in Figures 2, 3, 6 and 7) and hence a much lighter overburden. Finally, the values measured are for snow and definitely not for air. Measurements in air are almost always very easy to recognize: heating curves are erratic because of the complex and irregular convection that always takes place. This was alluded to in the methods section (lines 127-130) and we now further stress the point line 230 "It is fairly certain that this NP was not in an air gap, because measurements in air almost always produce erratic heating curves due to complex convection, and none of the heating curves were suspicious." We therefore believe that, even though there are clearly spatial variations, our values are representative and this is supported by the field data of Figures 2 and 7, with values similar to the automatic measurements.*

The inclusion of the soil in the interpretation is very useful, except that no detailed granulometric soil analysis seems to exists as this is a well investigated research site? More detailed data would clarify the observed behavior of the soil-freezing behavior. In fact, the observed curve indicates that the soil is not a silt, but a fine sand.

*This is an excellent point. Granulometric data was indeed lacking. We performed such an analysis using a Horiba partica LA-950V2 laser scattering particle size analyser. Data show a bimodal size distribution with modes centered at 17 and 59 µm. If the standard 50µm size limit between sand and silt is used, then our sample is 65% silt and 35% sand by mass. A subsection was added to our methods section to describe briefly the method (lines 171-174). The results are now mentioned lines 258-261.*

The authors put substantial weight on the effect of water vapor fluxes on the snowcover. The explanation of the fragile depth hoar bottom layer, as well as the formation of indurated layers, is based on the interpretation of temperature and vapor pressure gradients. The calculation of the vapor flux is omitted with the argument that the diffusivity is not well known. Laboratory experiments and numerical simulations (Calonne, 2014; Pinzer, 2012) defined the diffusion

coefficient precisely - in fact, due to the hand-to-hand process, the diffusivity in air is a very precise approximation. Approximate calculation for the season 2014-2015, with an average snow temperature of -30 deg C, temperature gradient 50 K m-1, and a duration of 90 days, result in a mass flux of 0.24 kg m-2. This flux seems to me too small to explain the observed processes.

*Thank you for this very useful comment and for taking the time to make the calculation. Using the data of Calonne et al., we have calculated water vapor fluxes and these are now shown in Figure 12. Our calculations lead to a total mass loss about 10 times greater than the Reviewer's estimate, because he used a temperature of -30°C while in the lower part of the snowpack, especially in early season, the temperature was around -5°C, so that the water vapor pressure was about 10 times greater. In any case, we agree that this is not quite sufficient to fully explain the total snow collapse. We therefore now suggest that diffusive fluxes alone cannot explain our observations, but convection and air advection (wind pumping) probably also took place and must have contributed to the mass loss, explaining the observation. The presence of convection cells is fully consistent with our observation of irregular collapse (Figure 8). This is now detailed in the discussion, lines 381-395.*

Obviously, spatial variability of the thermal conductivity of snow cannot be measured by permanent stations. However, as is obvious from the snow profile and the descriptions, spatial variability is an issue at the dm - m scale. As a suggestion, long snow profiles as done by Rutter et al (2014) in the arctic, or as demonstrated using a penetrometer by Proksch et al., would have contributed much to reduce the uncertainty in the measured values and their interpretation.

*Indeed, more measurement would have been highly desirable, as always, but there is only so much we can do in a 10-day campaign with complex logistics, multiple objectives and occasionally uncooperative weather. We did however investigate the spatial variability of the density of the depth hoar layer as shown in Figure 2. Figure 8 a and b also illustrate this variability.*

The use of needle probes as monitoring devices is strongly defended by the authors. However, a careful inspection of their Fig. 2 and Fig. 13 a) and calculating thermal conductivity based on the well accepted Calonne et al (2011) parameterization (or the Yen-parameterization) using the measured density, shows that the needle probes underestimate severely (for depth hoar a factor of about five) the effective thermal conductivity.

*Thank you for this interesting comment; however we have to disagree with the reviewer. First of all, we do not "strongly" defend the use of the heated needle probe. We are fully aware of its limitations and artefacts (thanks in part to the reviewer's publications, by the way). We only state that today it is the only technology suitable for the continuous monitoring of snow thermal conductivity, but we'll gladly consider any alternative, when available. Second, we do not think that the parameterizations of either Calonne et al. or Yen et al. are well accepted. The parameterization of Calonne is based on 30 values and that of Yen on less than 60. Calonne et al. only measured Alpine snows and did not measure a single sample of low-density depth hoar. Most or all of the snows used by Yen are not Arctic snow. Given the huge difference between Arctic and Alpine or*

*temperate snow, of which most snow scientists are not fully aware, we feel that it is not reasonable to attempt to determine the thermal conductivity of Arctic snow from very small and not representative data sets. Calonne et al., on their Figure 1, also report the 500+ values measured by (Sturm et al., 1997), all of them on Arctic and subarctic snow. These clearly show the huge difference between (sub)Arctic snow and Alpine snow, and demonstrates beyond doubt that parameterizations not developed for Arctic snow simply should not be used for Arctic snow. The large data set of (Sturm et al., 1997) also shows that for a given density, the range of thermal conductivities varies by a factor of 5, so that density correlations cannot predict accurately Arctic snow thermal conductivity. We admit that needle probes may underestimate the thermal conductivity of depth hoar, and we did mention that in our paper, but this effect is simply not sufficient to warrant the use of the parametrization of Calonne et al. or Yen to understand Arctic snow thermal conductivity. We have added a paragraph to our discussion (lines 467-476) to detail all this.*

The numerical simulation using Crocus seems to have major problems with creating a realistic density profile. As no details are given, my conclusion is that severe deficiencies must exist in the model parameterization. I suggest that the model runs are checked by an expert, as they seem to me beyond any reasonable behavior, or this part of the manuscript should be deleted.

*Full details of the model are given in (Domine et al., 2016), so we do not feel it is useful to repeat them here. The model runs were in fact performed for that earlier paper, of which Samuel Morin is a co-author, and he has an extensive record of publications using Crocus. The problem is not the model parameterizations or our use of the model, the problem is that the model has been developed for Alpine snow and does not take into account vertical water vapor fluxes. To fully convince the reviewer that this is the problem, we asked our colleagues Alexandre Langlois and Jean-Benoit Madore (University of Sherbrooke) to perform runs with SNOWPACK, the detailed snow physics model of the Reviewer's institution. They used NARR forcing data. Figure 1 below shows data similar to those of Figure 13 in our paper, which we also show here to facilitate comparison.*

[Figure]

*Figure 1. Simulations of snow thermal conductivity at Bylot Island using the SNOWPACK and Crocus models.*

*Neither Crocus nor Snowpack can reproduce the low thermal conductivity at the base of the Arctic snowpack and the high values at the top. We hope this will convince the Reviewer (and the Editor) that the problem is not our use of the model. The problem is that current detailed snow physics models (which include SNOWPACK and Crocus) miss what is arguably the most important process in Arctic snow: the upward water vapor transport due to the huge temperature gradient. Until this process is included in the models, sensible simulations of Arctic snow physics will not be possible. FD has been trying to get the message across for years, but most Alpine snow scientists (and that seems to be most snow scientists) do not seem to realize the importance of this process. We hope these results will help…*

——————————————————— **Technical corrections**

l 200 These values are questionable based on the authors density measurements. If there is no heat conducting matrix, there is no mechanical (compressive) strength.

*We believe we have discussed this in depth above. Again, density-thermal conductivity measurements cannot give accurate estimates.*

l 230 Did the authors any calibration of the temperature and soil humidity sensors before or after the deployment?

*In the methods section, line 91-92, we have added that "Water content sensors used the manufacturer's calibration for mineral soils and were not recalibrated, which may produce an error of up to 3%."*

l 283 "Rise" -> rise

*Corrected, thank you.*

l 290 The same limitations concerning vapor flux are valid also for convection (if there is any with the measured snow profile). In my view the speculation is out of place.

*As discussed above, water vapor fluxes are 10 times greater than estimated by the reviewer. Convection effects are also probably greater.*

l 317 ff The fluxes are easy calculate, this section should be rewritten in view of the actual fluxes.

*Indeed. The fluxes have now been calculated and Figure 12 changed accordingly. This led to what we feel is a significant improvement. Thank you for your comments of this aspect.*

Almost all Figures: The time axis is lettered in French, not English

*Changed.*

Fig. 1 The appearance of the vegetation in the photo seems to involve some vertical structure, completely flattened out during early winter? Not unimportant for the interpretation of the depth hoar formation.

*We now mention (lines 78-79) that "Vegetation consists of sedges, graminoids and mosses". In the results section, we also mention "Vegetation was observed to be mostly flattened by snow, with some sedge or graminoids stems still upright, but they did not seem to have impacted snow structure." lines 188-9.*

Fig. 2 The symbol for melt-freeze indurated depth hoar is actually defined (Int. Class., p. 19, a lying "8" with depth hoar symbols inside)

*We could not find this in the classification on p. 19 or elsewhere. The lying 8 on p.19 has circles inside. The word "indurated" is mentioned only once in the whole document, in the footnote of p. 17 and there is no associated symbol.*

Fig. 3 Snow depth or snow height. Caption, text and axis are not consistent (also Fig. 6)

*Thank you. For snow, we now use height throughout. We use depth for soil.*

Caption Fig 3: where there no easy measurements of snow depths around the stations to know spatial variability around?

*Sure, we apologize for failing to detail this. We have now added line 202 that "Measurements using an avalanche probe at 236 spots within 200 m of our site on 12 May 2015 showed a mean snow height of 25.3 cm, with a standard deviation of 13.1 cm". We make a similar statement on line 264 for the 2014 season.*

Fig. 7 The measured thermal conductivity data are inconsistent with the density profile. Give error bars.

*We hope that the above discussion will convince the reviewer that they are perfectly consistent. This is Arctic snow, not Alpine snow. Errors on measurements are detailed in the text. For example, in line 227, we specify that "the low value can be attributed to a negative systematic error of about 20% caused by the NP method, as described in (Riche and Schneebeli, 2013) and discussed above". Adding error bars would be confusing, as these are normally used for statistical, not systematic errors.*

**Acknowledgement**
We thank Jean-Benoit Madore and Alexander Langlois, University of Sherbrooke, QC, Canada, for making the SNOWPACK simulations of Figure 1.

**References**
Domine, F., Barrere, M., and Morin, S.: The growth of shrubs on high Arctic tundra at Bylot Island: impact on snow physical properties and permafrost thermal regime, Biogeosciences Discuss., 2016, 1-28, 2016.

Fauteux, D., Gauthier, G., and Berteaux, D.: Top-down limitation of lemmings revealed by experimental reduction of predators, Ecology, in press. in press.

Gauthier, G., Berteaux, D., Bety, J., Tarroux, A., Therrien, J. F., McKinnon, L., Legagneux, P., and Cadieux, M. C.: The tundra food web of Bylot Island in a changing climate and the role of exchanges between ecosystems, Ecoscience, 18, 223-235, 2011.

Riche, F. and Schneebeli, M.: Thermal conductivity of snow measured by three independent methods and anisotropy considerations, The Cryosphere, 7, 217-227, 2013.

Sturm, M., Holmgren, J., Konig, M., and Morris, K.: The thermal conductivity of seasonal snow, J. Glaciol., 43, 26-41, 1997.

---

## Author Comment (AC3) · 2 Sep 2016

**Reply to Matthew Sturm**

The Reviewer's comments are in black, and our response is embedded in the text, in blue italics. Line numbers refer to those of the version in track changes mode.

In this paper the authors report on the evolution of the thermal conductivity ( $k_{eff}$ ) of the tundra snow cover of Bylot Island. It is an interesting paper, with a key finding being that the contrast in  $k_{eff}$  between depth hoar and wind slab is on the order of 1:10. While this finding is not new, it is a useful piece of information that is neither widely appreciated nor captured by snow models (as the authors show). Overall, I recommend the paper be published....but not until it has been shortened considerably. The number of readers who will want to wade through all the detail currently in the paper in order to glean the main points is limited, and as consequence, the impact of the paper will be reduced

Thank you for this overall positive comment. Time will tell what the impact of the paper will be. The paper does not just present snow data but also original soil data, which are currently very scarce in the literature. Furthermore existing monitoring data on soil thermal conductivity arguably suffer from artefacts. Here, by minimizing the heating power of our needle probes, we argued that we limited some artefacts, and we feel that this adds interest to our data. Regarding the length of our paper, we feel that establishing links between meteorological events and the evolution of the thermal conductivity of snow deserves detailed discussions. For example, a detailed understanding between wind speed and wind slab physical properties such as thermal conductivity is critically needed. Current parameterization of wind effects is based on limited knowledge (see e.g. (Vionnet et al., 2012)) and at least some modellers will doubtless appreciate these details: some want to move away from simple and inaccurate density-thermal conductivity correlations. Lastly, the reviewer, while recommending considerable condensation, also requests numerous very lengthy additions. Doing both does not seem easy...

One other key point: if the 1:10 contrast in  $k_{eff}$  is the big take-away message of the paper, then the authors have missed an opportunity to discuss a key aspect of how tundra snow functions, in a way that is convolved with this 1:10 contrast. As seen in the authors' Figures 2 and 7, the relative percentage of depth hoar vs. wind slab controls the bulk thermal conductivity of tundra snow covers to a large extent. The number of windstorms, the relief of the underlying tundra tussocks (inter-tussock depth hoar), and the sequence of snowfall events all combine to determine what these percentages will be, hence how conductive the snow cover will be. This is a sensitive balance, one that could easily change if the climate, wind regimes, and snow-up dates, change. Discussing this would be valuable and for those readers that this paper aims at possible new information.

Such a discussion would deserve a paper on its own, and in any case this is not compatible with the considerable text condensation recommended above. We nevertheless address this issue in section 4.1 of the discussion, and mention how

threshold effects will lead to the abrupt transition between depth hoar and wind slab (lines 375-379). Thresholds may be reached by a sudden snow accumulation episode, whether caused by precipitation or wind. Note that the Reviewer stresses the tussock-type topography, which is frequently encountered on the Alaska North slope where he has done extensive work. This does not exist here, although hummock microrelief exists but not at our very site, and we therefore cannot discuss that here.

**Other specific comments follow:**

1. The authors have a tendency to cite new rather than seminal work, and philosophically, I find this distressing. It leads to the field "forgetting" facts and findings, then these have to be rediscovered. For example, that there are large amounts of organic matter in the permafrost was well known and reported on in the 1960s and 1970s: see Lachenbruch and Ferrians among other authors. A similar comment on references in the snow literature.

We agree with the Reviewer that we do not mention many seminal references. However, for the sake of concision, this was intentional and there are several reasons for this. For example, in the case of organic matter in the permafrost, we feel that this aspect, although important for many purposes, is not critical for our work, and we therefore do not wish to review it, even shortly. We therefore feel that a single recent (and therefore more complete) reference (Hugelius et al., 2014) is sufficient to inform the reader on the subject. Readers interested in more detail can look through the very many references of that paper. Regarding seminal snow references, we agree that there are many seminal references relevant to this work such as the work of Akitaya and in particular (Akitaya, 1975). However, again, our purpose is not to review depth hoar studies. In fact, Akitaya's work is not very relevant to Arctic depth hoar, as the Reviewer subsequently mentions, and we prefer to cite more relevant work, which by the way we find just as seminal, e.g. (Sturm and Benson, 2004; Sturm and Benson, 1997). And again, we are urged to remain concise and try to do so. Regarding soil thermal conductivity, we do cite seminal work, such as (Penner, 1970), line 138.

2. Line 117: Convection in snow. This was a lot of attention paid this topic in the 1980s and 1990s. The authors should see and cite:

Powers D, O'Neill K, Colbeck SC (1985) Theory of natural convection in snow. Journal of Geophysical Research 90:10641-10649.

Sturm M, Johnson JB (1991) Natural convection in the subarctic snow cover. Journal of Geophysical Research 96 (B7):11657-11671.

Sturm M (1991) The Role of Thermal Convection in Heat and Mass Transport in the Subarctic Snow Cover. USA-CRREL Report 91-19.

Indeed, we are well aware of these references. However, those papers deal with natural convection in the snow cover. Here, we only briefly discuss convection induced by the

heating of the needle, which in this case causes an artefact. We therefore believe that natural convection is not directly relevant to this discussion and again, to remain concise, we feel that it is not essential to discuss it here.

**In response to a comment by Reviewer 3, we invoked natural convection to explain the irregular collapse of depth hoar and cite (Sturm and Johnson, 1991), line 387.**

3. Line 151: Alternate methods of measuring thermal conductivity: It is not clear that guarded hotplate methods would produce more accurate or appropriate values  $k_{eff.}$  ---just different. While the dynamic method of the needle probe has some problems (pointed out by the authors) so too do steady-state methods. For example, real snow covers rarely, if ever, develop a steady-state non-varying temperature gradient. Also, when snow is subjected to uni-directional gradients, the thermal conductivity evolves through metamorphism, leading to a varying value of  $k_{eff}$  (or alternately, such mild gradients have to be imposed that the resultant tests are equivocal). I suggest altering the statement to indicate that the alternate methods are impractical, and *may not even be more accurate*.

This is currently a very controversial topic. Our understanding of this issue differs from that of the Reviewer. Clearly, it would be in our interest to be able to state that needle probes have no disadvantage relative to other methods. However, we feel that the recent work of (Calonne et al., 2011) and of (Riche and Schneebeli, 2013), which we cite in our paper, fairly convincingly point to a systematic error in needle probe measurements. We discuss this in detail in our earlier work (Domine et al., 2015). We briefly reminded the main important points of our earlier discussion lines 142 to 161. The Reviewer points to possible problems with steady state methods, and he is certainly correct: no current method to measure snow thermal conductivity is fully satisfactory, but we feel that current evidence does point to more problems with the needle probe method, which we must mention for full information of the reader. Fig. 6 of (Riche and Schneebeli, 2013) nicely sums up some aspects of the problem. By the way, we are not sure to understand the argument that natural snow covers do not develop a steady state temperature gradient. Perhaps the Reviewer is suggesting that steady state method may be used for thermal conductivity monitoring, but we probably all agree that this is not possible.

4. Line 156: Do you have to worry about pore water migration in measuring thermal conductivity of the soil using a needle probe?

This would be the equivalent of convection when measuring thermal conductivity in snow. We have never seen this aspect mentioned and do not expect it to be important given the very low permeability of soils. In any case, it is certainly much less important that problems due to melting, which we discuss in detail in section 4.4.

5. Lines 185 and 265-270: *Indurated depth hoar*. I would like to see a more comprehensive discussion of this snow texture. I believe the lead author and I discussed this snow texture when working jointly on snowpits in Alaska about a decade ago. He is correct that no formal symbol for the material exists in the International Classification,

but that arises in part from the fact that I don't think the committee charged with revising the classification believed such a material existed. The symbol I have been using for almost 20 years for this type of snow is a combined symbol of wind slab (black circle with a slash) and depth hoar (chains of grains or just cups). This is not that different than the one introduced in the paper, and it would be good to mention both as many of my students continue to use our older symbol. In addition, Carl Benson and I described this texture in our 1993 paper the phenomenon, though it was a few years later I introduced the term "indurated":

Elsewhere wind slabs are adjacent, one on top of the other. As the winter progresses even dense wind slabs can begin to metamorphose into depth hoar. We have observed wind slabs as dense as 0.35 g cm metamorphose into depth hoar by the end of the season.

Before introducing that term I corresponded with Dr. Akitaya (arguably at the time *the* expert on depth hoar) regarding it. He had never seen this texture in Japan because the temperature gradients are too low to produce it.

Indeed, FD does remember well this March 2004 discussion with the Reviewer near Barrow, Alaska. Indurated depth hoar is frequent in the Arctic and its absence in the international classification is a very big problem. FD mentioned this to Charles Fierz when he circulated drafts of the classification, but that steered no interest in the committee. We therefore fully agree that many world snow experts are familiar with midlatitude snow, but have seldom seen Arctic snow, even though it is more prevalent. There is therefore a need to fully describe indurated depth hoar. We have done that lines 189-201, quoting the most detailed description we found, from (Derksen et al., 2009) and (Domine et al., 2016). We also discuss that indurated depth hoar can form not only in wind slabs but also in refrozen layers, and propose 2 symbols to differentiate both types. Additional details are given lines 302-308. We also mention the works of (Hall et al., 1991) and of (Sturm et al., 2002), who mentioned a snow type that was presumably depth hoar, although not named specifically. We also mention (lines 303-309) the symbols used by those authors but we feel the symbols we propose are more logical and above all allow differentiating whether indurated depth hoar formed in wind slabs or melt-freeze layers

6. **Temperature Gradients**: (Lines 313-317) In order to turn wind slabs into indurated depth hoar, strong gradients are needed (see authors Figure 11). While the gradients a Bylot Island are super-critical (> 25°C/m) for much of the winter, they are less strong than I would have expected. I attach comparable data from Alaska's North Slope. Not only are the gradients stronger than those in the paper, but they start earlier in the winter. In a second graph, I have smoothed the data (48 hour running average), as the authors have done. The result is noticeably milder. I suggest the following changes/additions: a) don't smooth the gradients. In 48 hours a lot of metamorphism can take place. Smoothing makes the graph cleaner but less "physical" as far as metamorphism is concerned, b) show the critical gradient....whether that of Marbouty or Colbeck or Armstrong. It gives readers an appreciation of what drives the changes, c) consider computing something like an integrated metric called "time under a strong

gradient" or the like. Right now the paper mainly focuses on the agreement between the thermistor and NP values, but that misses the BIG point of what is actually driving the physics. And while Lines 325-330 provide the vapor gradient (and state that it can grow depth hoar), the values are really meaningless unless they are translated into something like a growth rate. Consider as an example computing how fast these vapor gradients could grow a thin sheet of ice...like the skirt on depth hoar cup. That would make the values mean something.

We have replaced the smoothed by the unsmoothed gradient in Figure 11. Values are a little bit greater, but not much: the highest value is 104 K m-1 vs. 87 for smoothed values. Gradient values are then just lower than in Alaska 25 years earlier. Higher values were probably reached earlier in the season but we need more than 12 cm of snow to measure it, since this is the level of our second NP. Our thermistor at 7 cm height would have given us the early season data, but as mentioned in the paper, the cable was chewed by a fox. This is detailed lines 355-361.

Following also the recommendation of Reviewer 3, we have calculated the water vapor flux, which is in fact what drives metamorphism, rather than the temperature gradient. We discuss threshold values and their relation to snow accumulation (lines 376-380). However, we do not wish to insist on "critical gradients" as we believe these have little physical meaning for Arctic snow. The real important physical variable is the water vapor flux, and we focus on that (Figure 12 and lines 366 ff.). The "critical gradient" was meaningful for Alpine and temperate snow, when the warmer temperature of the gradient was always close to 0°C. In Arctic snow, the warmer temperature can be as low as -35°C (Figure 3) so that the flux here, for a similar gradient, will be 15 times lower than if the warmer temperature were 0°C. We therefore think that reasoning on gradients for Arctic snow is misleading and we do not wish to follow that avenue.

**8. **Soil to Snow Fluxes**: These have been measured during depth hoar metamorphism. See:**

Trabant D, Benson CS (1972) Field experiments on the development of depth hoar. *Geological Society of America Memoir* 135::309-322.

Santeford HS Snow Soil Interactions in Interior Alaska. in Modeling of Snow Cover Runoff, USACRREL, Hanover, NH, pp. 311-318.

Sturm M, Benson CS (1997) Vapor transport, grain growth and depth hoar development in the subarctic snow. Journal of Glaciology 43:42-59.

We mention (Sturm and Benson, 1997) line 409 and compare our soil water loss value to theirs line 413. The other 2 references could not be found.

9. Line 341: There is a large literature on depth hoar and subnivean animals. Possibly the best starting point is W. O. Pruitt's 1957 paper:
Pruitt Jr, W. O. (1957). Observations on the bioclimate of some taiga mammals. *Arctic*, *10*(3), 130- 138.

Indeed there is, but further developing this aspect is not compatible with condensing the manuscript and with Reviewer 3's recommendation to even remove this section. We therefore limit ourselves to citing the work done at our research site, adding the recent results of (Fauteux et al., in press) which allow us to relate in a very preliminary manner our observed snow properties with lemming populations (lines 433-438).

**REFERENCES**

Akitaya, E.: Studies on depth hoar. In: Snow Mechanics (Proceedings of a symposium held at Grindelwald, April 1974) IAHS Pub. 114, Nye, J. (Ed.), 1975.

Calonne, N., Flin, F., Morin, S., Lesaffre, B., du Roscoat, S. R., and Geindreau, C.: Numerical and experimental investigations of the effective thermal conductivity of snow, Geophys. Res. Lett., 38, L23501, 2011.

Derksen, C., Silis, A., Sturm, M., Holmgren, J., Liston, G. E., Huntington, H., and Solie, D.: Northwest Territories and Nunavut Snow Characteristics from a Subarctic Traverse: Implications for Passive Microwave Remote Sensing, J. Hydrometeorol., 10, 448-463, 2009.

Domine, F., Barrere, M., and Morin, S.: The growth of shrubs on high Arctic tundra at Bylot Island: impact on snow physical properties and permafrost thermal regime, Biogeosciences Discuss., 2016, 1-28, 2016.

Domine, F., Barrere, M., Sarrazin, D., Morin, S., and Arnaud, L.: Automatic monitoring of the effective thermal conductivity of snow in a low-Arctic shrub tundra, The Cryosphere, 9, 1265-1276, 2015.

Fauteux, D., Gauthier, G., and Berteaux, D.: Top-down limitation of lemmings revealed by experimental reduction of predators, Ecology, in press. in press.

Hall, D. K., Sturm, M., Benson, C. S., Chang, A. T. C., Foster, J. L., Garbeil, H., and Chacho, E.: Passive microwave remote and insitu measurements of Arctic and sub-arctic snow covers in Alaska, Remote Sens. Environ., 38, 161-172, 1991.

Hugelius, G., Strauss, J., Zubrzycki, S., Harden, J. W., Schuur, E. A. G., Ping, C. L., Schirrmeister, L., Grosse, G., Michaelson, G. J., Koven, C. D., O'Donnell, J. A., Elberling, B., Mishra, U., Camill, P., Yu, Z., Palmtag, J., and Kuhry, P.: Estimated stocks of circumpolar permafrost carbon with

quantified uncertainty ranges and identified data gaps, Biogeosciences, 11, 6573-6593, 2014. Penner, E.: Thermal conductivity of frozen soils, Can. J. Earth Sci., 7, 982-987, 1970.

Riche, F. and Schneebeli, M.: Thermal conductivity of snow measured by three independent methods and anisotropy considerations, The Cryosphere, 7, 217-227, 2013.

Sturm, M. and Benson, C.: Scales of spatial heterogeneity for perennial and seasonal snow layers, Annals of Glaciology, Vol 38, 2004, 38, 253-260, 2004.

Sturm, M. and Benson, C. S.: Vapor transport, grain growth and depth-hoar development in the subarctic snow, J. Glaciol., 43, 42-59, 1997.

Sturm, M., Holmgren, J., and Perovich, D. K.: Winter snow cover on the sea ice of the Arctic Ocean at the Surface Heat Budget of the Arctic Ocean (SHEBA): Temporal evolution and spatial variability, J. Geophys. Res., 107, 8047, 2002.

Sturm, M. and Johnson, J. B.: Natural-convection in the sub-arctic snow cover, Journal of Geophysical Research-Solid Earth and Planets, 96, 11657-11671, 1991.
Vionnet, V., Brun, E., Morin, S., Boone, A., Faroux, S., Le Moigne, P., Martin, E., and Willemet, J. M.: The detailed snowpack scheme Crocus and its implementation in SURFEX v7.2, Geosci. Model Dev., 5, 773-791, 2012.

---

## Author Response (AR2)

**Responses to the Reviewers' comments**

Our responses are in blue italics, embedded into the reviewers' comments. Line numbers refer to the document in track changes mode.

5      **Response to M. Schneebeli**

1: Dr. Schneebeli is satisfied with your responses to his comments but has provided some suggestions for consideration in the final version of the paper:

- Concerning subnivean life:

Actually, subnivean life is very active also in alpine snowpacks. There is a lot of activity (especially mice) under alpine

10     snowpacks. Section 4.2 is still unexpected and in my view not relevant for this paper.

*Yes indeed, there are active small mammals under Alpine snowpacks. FD is well aware of that through the work of his old school buddy Nigel Yoccoz (see e.g. (Yoccoz and Mesnager, 1998)). However, the peculiarity of lemmings is that they reproduce under the snow, while Alpine small mammals do not. But since this is a paper on Arctic snow and this is an ancillary topic, we feel that we do not need to mention mice in Alpine snow. In any*

15     *case, we have revised and condensed section 4.2 (Snowpack structure and subnivean life) according to the editor's suggestion.*

- Concerning the thermal conductivity of depth hoar:

Concerning the compactive strength of depth hoar, indeed the compilation by Shapiro et al (Fig. B5) suggests that depth hoar with a density of 300 kg m-3 easily supports the weight of the snowpack studied (about 0.6 kPa). If this is the case, then the

20     mechanical stability contradicts the very low thermal conductivity, as these processes are in an intimate physical relationship. I suggest that the authors also consider the parameterization by Löwe et al. 2013, eq. 5, and fig. 1. This dataset includes more data points (167) than the paper by Calonne et al, and also at ice fraction below 0.1 (i.e. a density below 90 kg m-3). I would also like to know how the authors conclude that their (single) point value is representative, but at the same time write "even though there are clearly spatial variations". I am not able to extract any uncertainties from their figs. 2 and 7, or ranges of

25     thermal conductivities. In summary, all current parameterizations suggest that the value measured at the bottom is highly uncertain, and needs further investigation.

*A lot of the Reviewer's arguments are based on density correlations of snow properties, which very much surprises us. It has been long well known that snow physical properties only loosely correlate with density. Therefore, basing any detailed quantitative argument on density correlation has in our opinion little interest. As*

30     *already mentioned in our first rebuttal, (Sturm et al., 1997) show in their density-thermal conductivity correlation using 500+ values, that for a given density, the thermal conductivity varied by a factor of five. The Reviewer bases his comment this time on (Shapiro et al., 1997), but those authors also state (page 2) "data on mechanical properties and deformational behavior have usually been organized and presented as functions of the snow density. However, we will show from the literature that snow density is not a reliable indicator of these*

35     *properties.". The Reviewer himself concludes to that in his work on specific surface area (Matzl and Schneebeli, 2006) (page 562): "The comparison between density and SSA (Fig. 4) shows a loose correlation between these properties, as high SSA values are usually related to low densities and vice versa. [...] This makes a direct correlation, as suggested by Narita (1971) and Legagneux and others (2002), very uncertain in this density range." More generally, mechanical and thermal properties are certainly correlated, as in fact demonstrated for*

40     *the first time by (Domine et al., 2011), but the correlation is not even as good as between thermal conductivity and density, as shown by (Domine et al., 2011), see their Figure 2. In conclusion, we feel that any detailed reasoning based only on correlations between physical properties has little interest and we do not wish to*

*pursue this. The work by Löwe et al. is very interesting but only uses a limited number of samples, most or all of them from Alpine-like snow. That they do not find any snow with low thermal conductivity is not surprising. In fact, their method to measure snow thermal conductivity (micro CT) requires sampling, while we did in-situ measurements. Our depth hoar samples could not be sampled, as most of them just fell apart at the slightest contact. Therefore, no low values can be obtained with the method used by Löwe et al. and their work cannot be used to derive conclusions on Arctic depth hoar. Instead, both (Sturm et al., 1997) and (Domine et al., 2011) , who measured Arctic snows, find low values in their compilations, with values below 0.03 W m$^{-1}$ K$^{1}$ for densities around 200 kg m$^{-3}$. We detail that lines 485-498.*

*We acknowledge, however, that the data presented in the previous version of our paper lacked representativity. We have therefore added Table 1 where we report 23 thermal conductivity and 28 density values for the basal depth hoar layer.*

*Regarding uncertainties on thermal conductivity measurements with heated needle probes, we mention clearly and in detail in the methods section (lines 150-155) that in a worst case scenario, the error is 29%.*

- Concerning water vapor fluxes:

I agree with the revised calculations of the authors. As the authors point out, an extended air gap is not visible outside of the collapsed void or burrows in Fig. 8. Convective cells would even reduce the vapor flux, as there would be more cooling of the soil, and a convective cell formation seems also improbable as the authors describe hard and consequently less permeable layers above the depth hoar layer (Sturm et al made their measurements in a taiga snowpack).

*Sure, but we invoke convection at the beginning of the season, i.e. before low permeability wind slabs form.*

- Concerning the mass loss, see also the paper by Ebner et al 2015, in this (and other) papers it is shown that advecting air is saturated within a few millimeter distance, and cannot contribute significantly to mass transport.

*Sure, but air in snow is always saturated anyway, or in any case the deviations from saturations are so small that they cannot be measured. That does not stop mass transfer. Advection will accelerate mass transfer by increasing the air flux.*

In this view, I suggest that the authors formulate alternate reasons for the formation of the air gaps at the bottom of the snowpack more openly. I suggest that reasons are:

- burrows by lemmings

*This cannot be the case as indicated clearly lines 313-314 and as illustrated in Figure 8. Telling collapsed featured from burrows is simple.*

- possibly spontaneous collapse of the depth hoar during snow pit excavation

*Indeed this did happen, but careful digging and observations showed that spontaneous collapse did take place. Moreover, spontaneous collapse has already been observed by automatic measurements (Domine et al., 2015), see their Figures 4 and 5, which did not involve digging. Spontaneous collapse just does happen, however surprising that may seem to the Reviewer.*

*In any case, to satisfy the Reviewer, we have mentioned that this point is not fully resolved (lines 413-414). Also please note that we do propose an alternate explanation not even mentioned by the Reviewer: vapor diffusion enhancement, a process postulated and evidenced decades ago (lines 410-413).*

- Concerning the conclusions, I believe that it should be mentioned that the data and the models are not completely conclusive, and additional measurements are necessary.

*Well, the problem is clearly the models. The data seem OK to us and to the other Reviewers.*

Please note that Martin Schneebeli indicated that he wished to remain anonymous in the Acknowledgements.

*Sure, we deleted his name from the acknowledgements. His review is signed, however, so this will look odd.*

85    Löwe, H., Riche, F. & Schneebeli, M. 2013 A general treatment of snow microstructure exemplified by an improved relation for thermal conductivity. Cryosph. 7, 1473–1480. (doi:10.5194/tc-7-1473-2013)

Ebner, P. P., Andreoli, C., Schneebeli, M. & Steinfeld, A. 2015 Tomography-based characterization of ice-air interface dynamics of temperature gradient snow metamorphism under advective conditions. J. Geophys. Res. Earth Surf. 120, 2437–2451. (doi:10.1002/2015JF003648)

90

**Response to M. Sturm**

2. Dr. Sturm is satisfied with the revised m/s but feels it could be improved further with more concise wording and attention to typographical errors. He asked that the following items be addressed before publication:

- There are a fair number of typos in the revised text.....the authors need to carefully clean these up., not pass them on to the

95    tc staff editors.

*We have checked our text carefully and apologize for any remaining error.*

- Indurated depth hoar: in my last review I reminded the lead author that I introduced him to this snow texture I think in about 2006(?). He cites our paper (Derksen et al., 2009) as the first published description, and as I recall, the section in that paper he quotes is a section I wrote. But I would request that he cite where this was first described in greater detail (attached -

100    note: copy of report e-mailed to lead author by ed.) :

Sturm, M., Derksen, C., Liston, G., Silis, A., Solie, D., Holmgren, J., & Huntington, H. (2008). A reconnaissance snow survey across Northwest Territories and Nunavut, Canada, April 2007 (No. ERDC/CRREL-TR-08-3). ENGINEERING RESEARCH AND DEVELOPMENT CENTER HANOVER NH COLD REGIONS RESEARCH AND ENGINEERING LAB.

105

In that easily obtained report, we describe indurated depth hoar on pages 6, 14, 18, and provide a percentage of the snow sampled in Figure 10.

*We have replaced the reference to Derksen et al. (2009) with that of Sturm et al. (2008) and used a quote from Sturm rather than from Derksen. (lines 201-209) This however does not result in any significant change to our*

110    *description.*

- A personal pique, but values like density and thermal conductivity we generally ascribe to a layer, and therefore (in lieu of an contradictory data) the same across the whole layer. Thus, when drawing a graph (Figures 2 and 7), they should be horizontal bars, rather than point-to-point lines. That was the old way it was done, and was more representative. It is easily done with most graphing programs.

115    *The Reviewer is making a good point. The CAAML format also recommends this in a number of cases and we agree this has many advantages. However, it is based on the old assumption that visual snow layers are homogeneous, while more and more studies show that this is often not the case (see e.g. (Matzl and Schneebeli, 2006)). Furthermore, it works well when layers are parallel and of constant thickness. In our specific case, it would also make it difficult to represent the thermal conductivity and density variations in the basal layer of*

120    *Figure 2. In Figure 7, given the huge short-scale variability, such a representation would not be possible. We realize that are plots are not ideal, but they are the best compromise we could find.*

- When citing references in the text, place them in date order (ascending or descending), and be consistent, please

*Sure, we adjusted EndNote settings.*

[revised manuscript text omitted]